# LLM Microscope: What Model Internals Reveal About Answer Correctness and Context Utilization

## Abstract

Although large language models (LLMs) have tremendous utility, trustworthiness is still a chief concern: models often generate incorrect information with high confidence. While contextual information can help guide generation, identifying when a query would benefit from retrieved context and assessing the effectiveness of that context remains challenging. In this work, we operationalize interpretability methods to ascertain whether we can predict the correctness of model outputs from the model's activations alone. We also explore whether model internals contain signals about the efficacy of external context. We consider correct, incorrect, and irrelevant context and introduce metrics to distinguish amongst them. Experiments on six different models reveal that a simple classifier trained on intermediate layer activations of the first output token can predict output correctness with about 75% accuracy, enabling early auditing. Our model-internals-based metric significantly outperforms prompting baselines at distinguishing between correct and incorrect context, guarding against inaccuracies introduced by polluted context. These findings offer a lens to better understand the underlying decision-making processes of LLMs.

## 1 Introduction

Large language models (LLMs) have shown tremendous utility in many domains, including those that require accurately answering factual queries (Scao et al., 2023; Grattafiori et al., 2024; Groeneveld et al., 2024). However, trustworthiness remains a chief concern: LLMs often generate convincing, but thoroughly incorrect and non-factual responses, termed *hallucinations* (Bang et al., 2023; Huang et al., 2025; Guerreiro et al., 2023).

Recently, retrieval-augmented generation (RAG) has been proposed to mitigate this problem (Lewis et al., 2020). Although RAG is effective, two challenges remain: confidence estimation to identify uncertain examples where an LLM requires external context and efficacy evaluation to score the utility of the retrieved external context. Confidence estimation is challenging as LLMs are poorly calibrated: models often assign high probabilities to incorrect generations, making it difficult to detect when retrieval is needed (Jiang et al., 2021; Mielke et al., 2022a; Kadavath et al., 2022a; Yin et al., 2023a). However, existing approaches either rely on fragile self-evaluation (Yin et al., 2023b; Chen et al., 2024) or focus on narrow tasks and require complex setups (Azaria & Mitchell, 2023; Burns et al., 2024).

For context evaluation, although there exist methods to estimate efficacy of retrieved context such as SelfRAG (Asai et al., 2023), they rely on fine-tuning models and prompting external models to gauge the utility of external context. A lightweight way to judge efficacy directly from model internals remains missing. Instead, we look at these questions using our *LLM microscope*, through the *lens of mechanistic interpretability*, and study whether model internals contain signals about the correctness of responses and the efficacy of specific auxiliary context when answering a query. Concretely, we study the following research questions:

1. **RQ1:** *Can we estimate the correctness of a model generation from its model internals alone?*
2. **RQ2:** *Can we estimate the efficacy of a given context directly from model internals?*

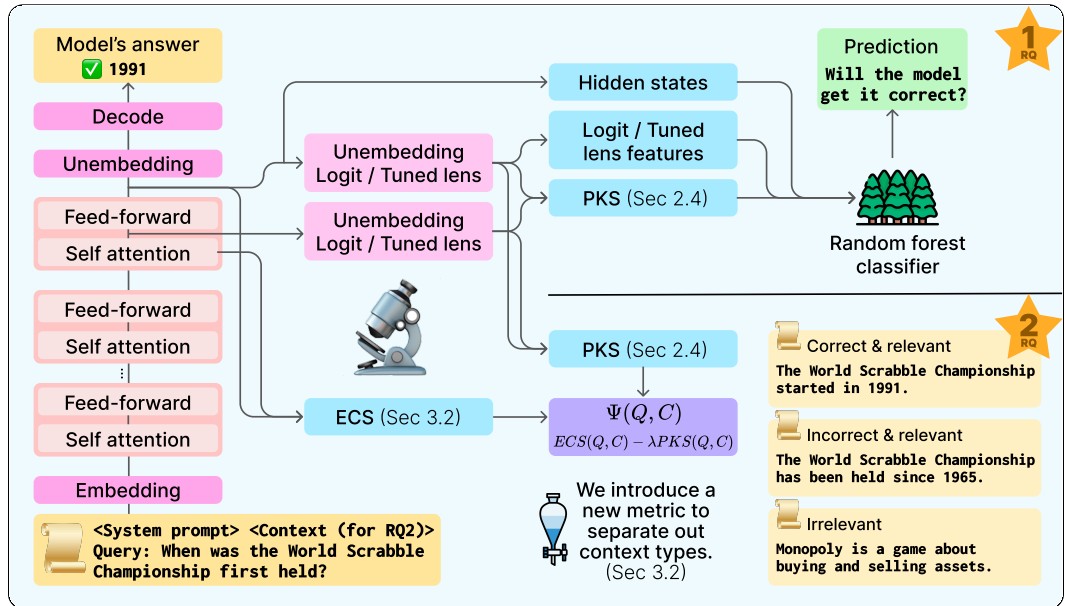

Figure 1: Overview of our framework. For RQ1, we use model internals, including hidden states, Logit/Tuned Lens-based features, and parametric knowledge score (PKS) to train classifiers that predict the correctness of a model's output when answering a question. For RQ2, we analyze how internal signals like external context score (ECS) and PKS respond to different types of external context (correct, incorrect, irrelevant) in order to assess the model's sensitivity to context when generating answers.

To answer these questions, we study six LLMs across three model families and sizes in an open-domain factual question-answering setting on the TriviaQA (Joshi et al., 2017) and MMLU (Hendrycks et al., 2021) datasets. We train classifiers on model internals to predict generation correctness and context relevance. To operationalize contextual relevance, we introduce two novel measures: *contextual log-likelihood gain* and *contextual relative utility* and study whether model internals can discriminate between contexts across two axes: correctness and relevance. We find that:

1. We can estimate correctness of a model generation to open-domain questions from model internals alone with over 75% accuracy and 70% AUC-ROC.
2. Using our model internals-based contextual log-likelihood gain metric, we can effectively discriminate between contexts across both the axes of correctness and relevance.

## 2 RELATED WORK

**Confidence Estimation**   Confidence estimation methods are typically categorized as *closed-box* or *open-box*. Closed-box methods prompt the model to assess its own correctness, either by estimating the likelihood that its answer is correct or by judging whether it knows the answer (Kadavath et al., 2022b; Tian et al., 2023; Yin et al., 2023b; Huang et al., 2024). Some use linguistic cues (Mielke et al., 2022b) or generation consistency (Manakul et al., 2023; Zhang et al., 2023; Chen et al., 2024). Open-box methods instead analyze the model's internals, either through logit-based uncertainty estimates (Murray & Chiang, 2018; Kadavath et al., 2022b; Huang et al., 2023; Vazhentsev et al., 2023) or by probing hidden activations directly (Li et al., 2023; Orgad et al., 2024; Burns et al., 2024; Subramani et al., 2025).

Our work builds on this line of open-box internal-state approaches but differs in several key ways. First, we focus on predicting the correctness of generated answers across both open-domain QA and multiple-choice settings (Azaria & Mitchell, 2023; Burns et al., 2024). Second, we evaluate whether a model's internals can predict the correctness of its own generations, establishing a direct, on-policy evaluation setting (Azaria & Mitchell, 2023; Servedio et al., 2025). We use a simple random

forest classifier to both preserve feature interpretability and avoid entangling effects of classifier complexity (Kadavath et al., 2022b; Orgad et al., 2024; Ashok & May, 2025). This simple yet effective setup allows us to demonstrate that a variety of internal-state features derived from diverse interpretability techniques can support robust confidence estimation in generative tasks. [1]

**Parametric and Contextual Knowledge** The activation spaces of LLMs encode structured, editable knowledge through mechanisms like induction heads (Elhage et al., 2021), knowledge neurons (Dai et al., 2022), and feed-forward layers (Geva et al., 2022). Probing tools such as Logit-Lens (nostalgebraist, 2020), Tuned-Lens (Belrose et al., 2023), Future-Lens (Pal et al., 2023), and Backward-Lens (Katz et al., 2024), and methods like causal tracing (Meng et al., 2022), attention analysis (Geva et al., 2023; Yuksekgonul et al., 2023), and steering (Subramani & Suresh, 2020; Subramani et al., 2022; Turner et al., 2023), reveal how predictions evolve across layers. Additionally, different layers of transformer-based language models encode different linguistic properties (Tenney et al., 2019; Ethayarajh, 2019; Li & Subramani, 2025). To improve factuality, retrieval-augmented approaches incorporate external context (Roller et al., 2020; Shuster et al., 2021; Ravichander et al., 2025). Recent work (Wu et al., 2024; Wadhwa et al., 2024; Sun et al., 2025; Li et al., 2025) examine clashes between internal and retrieved knowledge, identify model over-reliance on retrieved context, and develop metrics to diagnose and treat that reliance. We address this gap by moving beyond prompting-based approaches to context helpfulness measurement; to our knowledge, we are the first to apply interpretability techniques to study how models utilize contextual versus parametric knowledge (Huang et al., 2024).

## 3 RQ1: ESTIMATING CORRECTNESS

To study RQ1, we formulate a binary classification task to predict whether a generated answer is correct. Each instance in this task consists of a factual question $Q$, the ground-truth answer $A^*$, an LLM-generated answer $A$, and all activations and outputs $\mathcal{H}_{Q,A}$ produced by a model $M$ when attempting to answer $Q$. We train a simple classifier $f$ that takes $Q$, $A$, and $\mathcal{H}_{Q,A}$ as input and predicts $\mathbb{I}[A = A*]$. The classifier does not have access to the ground-truth answer $A*$ or any external knowledge source; $A*$ is only used to create the ground-truth label for training. We explore several choices for $\mathcal{H}_{Q,A}$ as input features, which we describe below.

### 3.1 ASKING LLMs DIRECTLY

We explore two prompting strategies to measure how well LLMs can express their confidence in an answer: *Prompting without Answers* and *Prompting with Answers*. In *Prompting without Answers*, the model $M$ is given a question $Q$, and asked to output a confidence score between 0 and 100 indicating how confident it is about answering the question without actually generating an answer. In *Prompting with Answers*, $M$ first generates a candidate answer $A$ for $Q$ and then is asked to output a confidence score based on how certain it is about the generated answer (see §B.1 for prompting setup details).

### 3.2 DECODING FROM INTERMEDIATE LAYERS

To test whether model internals contain signals for estimating answer correctness, we look at two techniques that facilitate decoding from intermediate hidden states: *Logit Lens* and *Tuned Lens* (nostalgebraist, 2020; Belrose et al., 2023). Both of these methods convert intermediate hidden states into the vocabulary space. We obtain the following input features $\mathcal{H}_{Q,A}$ from these probability distributions. These features are computed per layer and sequence position and provided at once to the classifier. See §B.2 and §B.3 for additional details on *Logit/Tuned Lens* and these features.

- The **Shannon's entropy** measures the uncertainty in the model's probability distribution over the next token at each layer and sequence position (Shannon, 1948).
- The **output token rank** is the position, in descending order of log-probability, of the token selected by the model at each layer. While this rank is related to probability, it serves as a more direct proxy for the token a decoding algorithm is likely to generate.

---

[1] See Geng et al. (2024) for a broader overview.

- The **top-$p$ presence** of the output token is a binary indicator of whether the token generated by the model appears in the top-$p$ nucleus set at each layer ($p \in \{0.5, 0.9, 0.95, 0.99\}$ Holtzman et al. (2019)).
- The **cross-entropy** quantifies how similar model predictions are across layers and is computed as the negative log probability of the generated token under the distribution at each layer.

### 3.3 HIDDEN STATES

Rather than transforming the hidden states and decoding from them, we experiment with using the model's hidden states directly as input features to $f$. Formally, we use $\mathbf{h} \in \mathbb{R}^d$ at layer $\ell \in \{1, ..., L\}$ from the first token position of the forward pass that generated $A$ in response to $Q$ to train $f$. Since activations could have different scales due to differences in parameter norms throughout training (Merrill et al., 2021), we convert the value in every dimension $d$ of the activation $h$ to a z-score using means and variances computed from an auxiliary dataset.

### 3.4 PARAMETRIC KNOWLEDGE SCORES (PKS)

We expect the amount of parametric knowledge used to answer a question $Q$ positively correlates with the confidence the model has about its generated answer $A$. In other words, the more parametric knowledge used to answer the question the more confidence it should have about that answer. To quantify the utilization of parametric knowledge, we use the *Parametric Knowledge Score (PKS)* from Sun et al. (2025), which measures how much the feedforward networks (FFN) contribute to the activations. For each token at layer $\ell$, the activations before and after the FFN layer are transformed into a probability distribution over the vocabulary via *Logit Lens*. PKS is the Jensen-Shannon divergence (JSD) between these two distributions, loosely capturing the amount of information imparted by FFN weights into the activations. Formally, the token-level PKS is given by $P_\ell = \text{JSD}\left(q(\mathbf{x}_{\text{before},\ell}) \,\|\, q(\mathbf{x}_{\text{after},\ell})\right)$, where $q(\cdot) = \text{softmax}(\text{LogitLens}(\cdot))$.

## 4 RQ2: ESTIMATING EFFICACY OF CONTEXT

Our goal is to see whether we can measure the efficacy of a given context $C$ from either internal (via model internals) or external (via prompting) features. We look at two attributes of context: correctness and relevance and define three types of context $C$:

- Correct and relevant ($C_{\text{correct}}$): aligns with the gold answer and has essential information.
- Incorrect but relevant ($C_{\text{incorrect}}$): structurally similar and topically aligned, but contains incorrect or misleading information.
- Irrelevant ($C_{\text{irrelevant}}$): topically unrelated to and unhelpful for answering the question.

We define two lenses with which we can observe the effect of contexts: *contextual log-likelihood gain* and *contextual relative utility*.

*Contextual log-likelihood gain* measures how much incorporating a context $C$ improves (positive) or degrades (negative) the model's confidence in generating the correct answer. In RAG, it quantifies the utility of the retrieved context to the model to generate the correct answer. For a question $Q$, context $C$, and ground-truth answer tokens $\mathbf{y} = (y_1, \ldots, y_T)$, the *contextual log-likelihood gain* is defined as:

$$\text{LL}(Q, C) = \sum_{t=1}^{T} \log p(y_t \mid y_{<t}, Q, C) - \sum_{t=1}^{T} \log p(y_t \mid y_{<t}, Q) \tag{1}$$

*Contextual relative utility* compares two different contexts $C_1$ and $C_2$ and measures whether $C_1$ is more helpful (positive) or harmful (negative) than $C_2$ for a model to produce the correct answer. We formally define *contextual relative utility* as $\Delta\text{LL}(Q, C_1, C_2) = \text{LL}(Q, C_1) - \text{LL}(Q, C_2)$.

### 4.1 PROMPTING-BASED CONFIDENCE ESTIMATION

We analyze whether prompting-based methods can approximate the model's contextual log-likelihood gain through asking the model to generate confidence scores. For each question $Q$, we prompt the

model to output a confidence score on a 0–100 scale under the two different context conditions discussed in §3.1: *Prompting without Answers* and *Prompting with Answers*.

Let $\text{Conf}(Q, C)$ denote the model's confidence score when answering $Q$ with context $C$, and $\text{Conf}(Q)$ denote its confidence when answering without any context. We define the *prompting-based contextual gain* as:

$$\Omega(Q, C) = \text{Conf}(Q, C) - \text{Conf}(Q). \tag{2}$$

To compare two contexts $C_1$ and $C_2$, we define the *prompting-based relative utility* as $\Delta\Omega(Q, C_1, C_2) = \Omega(Q, C_1) - \Omega(Q, C_2)$, which estimates the relative helpfulness of $C_1$ over $C_2$. If the model can accurately distinguish context quality, we expect: $\Delta\Omega(Q, C_{\text{correct}}, C_{\text{incorrect}}) > 0$, and $\Delta\Omega(Q, C_{\text{correct}}, C_{\text{irrelevant}}) > 0$.[2]

## 4.2 INTERNALS-BASED CONFIDENCE ESTIMATION

To measure how well additional context can affect question-answering ability along the two axes of external context utilization and internal parametric knowledge reliance, we use *external context score* (ECS) and *parametric knowledge score* (PKS) from Sun et al. (2025). *External Context Score (ECS)* captures the extent to which a language model relies on external context during generation. For each output token $t_n$, ECS is computed as the cosine similarity between the token's final-layer hidden state $x_n^L$ and the mean-pooled embedding $e$ of the top-$k\%$ most attended context tokens, selected based on attention weights. The token-level ECS is defined as:

$$\text{ECS}_h^l(t_n) = \frac{e \cdot x_n^L}{\|e\|\|x_n^L\|} \tag{3}$$

We can obtain the ECS score across a multi-token generation by simply averaging token-wise ECS scores over all output tokens. A higher ECS indicates stronger alignment between the generated output and the retrieved context, suggesting that the model is effectively utilizing external information.

In our setting, parametric knowledge and external context can be thought of as orthogonal. We define a proxy for *contextual log-likelihood gain* using model internals as:

$$\Psi(Q, C) = \text{ECS}(Q, C) - \lambda \cdot \text{PKS}(Q, C), \tag{4}$$

where $\text{ECS}(Q, C)$ measures the model's reliance on external context, $\text{PKS}(Q, C)$ quantifies the effect of parametric knowledge in the presence of context, and $\lambda$ is a scaling factor that rescales PKS to match ECS. In our experiment, we choose the value of $\lambda$ for each dataset such that the mean PKS score is rescaled to match the mean ECS score, computed across all examples and all context types. This normalization ensures that per-example differences in $\Psi(Q, C)$ reflect shifts in context reliance without introducing category-specific bias. To compare two contexts $C_1$ and $C_2$, we define *internals-based relative utility* as:

$$\Delta\Psi(Q, C_1, C_2) = \Psi(Q, C_1) - \Psi(Q, C_2) \tag{5}$$

Our formulation reflects the tradeoff between contextual and parametric knowledge: higher values of $\Psi(Q, C)$ suggest stronger reliance on external context, while lower values indicate dominance of parametric knowledge or potential confusion from misleading context.

In addition, we directly compute contextual relative utility using the actual log-likelihood, denoted as $\Delta\Psi^{(\text{log likelihood})}$. Specifically, we apply the log softmax over the vocabulary to obtain token-level log probabilities, average these over the ground-truth answer tokens, and then take the difference between two context types.

## 5 EXPERIMENTAL SETUP

**Datasets** We use factual short-question-answering datasets TriviaQA (Joshi et al., 2017) and MMLU (Hendrycks et al., 2021) for our experiments. We report results on both datasets for RQ1. As TriviaQA is open-ended while MMLU is multiple-choice, this ensures generalizability of our results across question types. As RQ2 requires accompanying context, we only use TriviaQA for this

---

[2]See §C.2 for the reasoning, and §C.1 for details on the prompting setup.

| Dataset | Context Type | LLaMA 3 8B | LLaMA 2 13B | LLaMA 2 7B | Gemma 2 9B | Qwen 2.5 7B | Qwen 2.5 3B |
|---|---|---|---|---|---|---|---|
| TriviaQA | None | 0.708 | 0.591 | 0.611 | 0.749 | 0.558 | 0.435 |
| | Correct | 0.888 | 0.833 | 0.845 | 0.906 | 0.856 | 0.815 |
| | Incorrect | 0.358 | 0.340 | 0.314 | 0.406 | 0.332 | 0.301 |
| | Irrelevant | 0.579 | 0.516 | 0.465 | 0.727 | 0.488 | 0.293 |
| MMLU | None | 0.611 | 0.484 | 0.462 | 0.709 | 0.693 | 0.634 |

Table 1: Model accuracies on TriviaQA and MMLU datasets with no context and with varying context conditions. MMLU does not contain context or evidence documents.

research question as MMLU does not provide context passages or evidence documents. To ensure our experiments remain computationally viable, we only consider the validation subset of TriviaQA, from which we retain 6,557 examples after quality filtering (see §D.1 for details). These are split 80%-20% for training and testing our classifiers. To account for possible variations in referring to the same entity, we use GPT-4o (OpenAI et al., 2024) as an LLM-judge to evaluate the correctness of model-generated answers against all the ground-truth answer variations present in TriviaQA. We use the 14,042 test examples of the "all" subset of MMLU as our training set and the 1,531 validation examples as our test set. As MMLU questions are multiple-choice, we use regex-based answer extraction and verification.

**External Context:** We use the original evidence document provided by TriviaQA, or a summarized version if the original exceeds 500 tokens, as the correct context. We then construct incorrect and irrelevant contexts by modifying or substituting these documents in controlled ways: for incorrect context, we ask an LLM to replace all references to the ground-truth answers with incorrect alternatives; for irrelevant context, we sample the correct context of a different example with low textual similarity (see §D.1 for details and examples).

**Models:** We experiment with 6 models across families and sizes: LLaMA-3-8B-Instruct (Grattafiori et al., 2024), LLaMA-2-7B-Chat-HF and LLaMA-2-13B-Chat-HF (Touvron et al., 2023), Qwen-2.5-3B-Chat and Qwen-2.5-7B-Chat (Yang et al., 2024), and Gemma-2-9B-It (Team et al., 2024).

**Methodology Details:** We choose random forest classifiers for their interpretability, which facilitates feature importance analysis, while also capturing non-linear patterns in the data. These classifiers are trained on the features discussed in §3 (see §D.2 for hyper-parameters and training speifications). For RQ2, we estimate *contextual log-likelihood gain* using ECS and PKS using equation (4). For ECS, we average across both attention heads and output tokens and for PKS, we average across output tokens to capture the overall effect of parametric knowledge on the entire output. PKS and ECS are calculated per layer and then averaged across layers. We present a layer-wise analysis in §7.2. We calculate one $\lambda$ for each model by rescaling the relative magnitudes of PKS and ECS, averaged over all examples and context types in the training set.

## 6 RESULTS

We present the accuracies of our six LLMs on both TriviaQA and MMLU in Table 1. All models achieve moderate accuracy in the default no-context setting, with the larger and more recent models generally performing better. As expected, accuracy on TriviaQA increases with correct context and drops with incorrect or irrelevant context, highlighting that models are indeed sensitive to external context. We focus on the no-context setting to study RQ1 in §6.1 and examine different context types for RQ2 in §6.2.

### 6.1 RQ1: CAN WE ESTIMATE CORRECTNESS FROM MODEL INTERNALS ALONE?

Table 2 shows that prompting-based baselines perform poorly across all models, barely outperforming the majority-class baseline, even when given access to its own generated answer. This is particularly evident in the LLaMA models, suggesting that such prompting strategies fail to estimate correctness accurately and highlight that LLMs are poorly calibrated.

| | Estimator | LLaMA 3 8B | LLaMA 2 13B | LLaMA 2 7B | Gemma 2 9B | Qwen 2.5 7B | Qwen 2.5 3B |
|---|---|---|---|---|---|---|---|
| **Accuracy** | Majority | 0.699 | 0.591 | 0.621 | 0.735 | 0.567 | 0.565 |
| | Prompt w/ A | **0.789** | 0.599 | 0.634 | 0.762 | 0.727 | 0.652 |
| | Prompt w/o A | 0.699 | 0.611 | 0.621 | 0.768 | 0.637 | 0.605 |
| | Logit lens | 0.782*‡ | **0.779***†‡ | **0.776***†‡ | 0.774* | 0.751*‡ | 0.729*†‡ |
| | Tuned lens | 0.779*‡ | 0.775*†‡ | 0.775*†‡ | - | - | - |
| | Hidden states (Best) | 0.759*‡ | 0.679*†‡ | 0.702*†‡ | **0.785*** | **0.782***†‡ | **0.749***†‡ |
| | PKS | 0.733 | 0.725*†‡ | 0.709*†‡ | 0.743 | 0.650* | 0.695*†‡ |
| **AUC-ROC** | Majority | 0.500 | 0.500 | 0.500 | 0.500 | 0.500 | 0.500 |
| | Prompt w/ A | 0.783 | 0.512 | 0.537 | 0.592 | 0.736 | 0.687 |
| | Prompt w/o A | 0.591 | 0.582 | 0.542 | 0.742 | 0.653 | 0.631 |
| | Logit lens | **0.790***‡ | **0.847***†‡ | **0.835***†‡ | **0.747***† | **0.826***†‡ | **0.812***†‡ |
| | Tuned lens | 0.782*‡ | 0.846*†‡ | 0.829*†‡ | - | - | - |
| | Hidden states (Best) | 0.647*‡ | 0.631*†‡ | 0.647*†‡ | 0.616* | 0.774*‡ | 0.735*†‡ |
| | PKS | 0.729*‡ | 0.768*†‡ | 0.743*†‡ | 0.723*† | 0.715*‡ | 0.752*†‡ |

(a) TriviaQA

| | Estimator | LLaMA 3 8B | LLaMA 2 13B | LLaMA 2 7B | Gemma 2 9B | Qwen 2.5 7B | Qwen 2.5 3B |
|---|---|---|---|---|---|---|---|
| **Accuracy** | Majority | 0.604 | 0.538 | 0.548 | 0.717 | 0.692 | 0.627 |
| | Prompt w/ A | 0.603 | 0.497 | 0.547 | 0.718 | 0.705 | 0.627 |
| | Prompt w/o A | 0.605 | 0.535 | 0.529 | 0.718 | 0.704 | 0.629 |
| | Logit lens | 0.705*†‡ | **0.692***†‡ | **0.699***†‡ | 0.769*†‡ | **0.815***†‡ | 0.673*†‡ |
| | Tuned lens | 0.695*†‡ | 0.671*†‡ | 0.684*†‡ | - | - | - |
| | Hidden states (Best) | **0.744***†‡ | 0.686*†‡ | 0.672*†‡ | **0.801***†‡ | 0.777*†‡ | **0.746***†‡ |
| | PKS | 0.605 | - | 0.543 | 0.705 | 0.691 | 0.618 |
| **AUC-ROC** | Majority | 0.500 | 0.500 | 0.500 | 0.500 | 0.500 | 0.500 |
| | Prompt w/ A | 0.590 | 0.501 | 0.560 | 0.557 | 0.562 | 0.629 |
| | Prompt w/o A | 0.497 | 0.538 | 0.492 | 0.513 | 0.544 | 0.519 |
| | Logit lens | **0.798***†‡ | **0.771***†‡ | **0.767***†‡ | **0.843***†‡ | **0.939***†‡ | 0.711*†‡ |
| | Tuned lens | 0.770*†‡ | 0.752*†‡ | 0.760*†‡ | - | - | - |
| | Hidden states (Best) | 0.740*†‡ | 0.684*†‡ | 0.576*‡ | 0.736*†‡ | 0.728*†‡ | **0.733***†‡ |
| | PKS | 0.537‡ | - | 0.543*‡ | 0.555*‡ | 0.541* | 0.538 |

(b) MMLU

Table 2: Performance of various classifiers on the test sets using our proposed methods and baseline approaches. We include two prompting baselines: Prompt w/ A (prompting with answers) and Prompt w/o A (prompting without answers) and a simple majority class baseline (Majority). For Logit Lens, Tuned Lens, and PKS methods, we use all values across layers as input features, while for Hidden States, we choose the best layer. Hidden states are normalized using z-score normalization. Tuned Lens results are omitted for models whose weights are not publicly available (Belrose et al., 2023). PKS scores are not available for LLaMA 2 13B on MMLU due to computational constraints. *, †, and ‡ indicate statistical significance compared to Majority, Prompt w/ A, and Prompt w/o A, respectively (p-value $< 0.05$, two-sided permutation test).

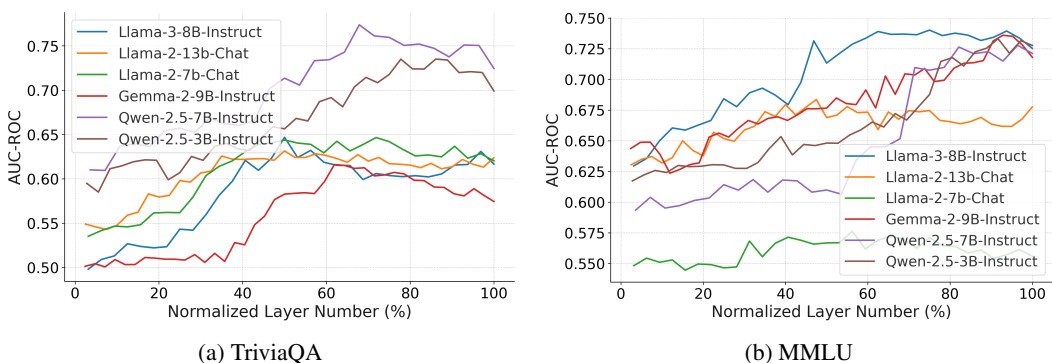

(a) TriviaQA          (b) MMLU

Figure 2: Area under ROC curve for random forest classifiers trained on z-score normalized hidden states of each layer. Performance increases with layer depth, suggesting that later layers refine and consolidate decision-relevant signals.

| Context Comparison | Differentiator | LLaMA 3 8B | LLaMA 2 7B | Qwen 2.5 7B | Qwen 2.5 3B | Average |
|---|---|---|---|---|---|---|
| Correct > Incorrect | $\Delta\Omega^{(\text{prompt w/ A})}$ | 19.7 | 10.9 | 32.7 | 16.4 | 19.9 |
| | $\Delta\Omega^{(\text{prompt w/o A})}$ | 20.8 | 11.3 | 30.0 | 23.6 | 21.4 |
| | $\Delta\Psi^{(\text{log likelihood})}$ | 83.9$^{\dagger\ddagger}$ | **89.5$^{\dagger\ddagger}$** | 82.6$^{\dagger\ddagger}$ | **83.3$^{\dagger\ddagger}$** | **84.8$^{\dagger\ddagger}$** |
| | $\Delta\Psi^{(\text{model internals})}$ | **85.5$^{\dagger\ddagger}$** | 70.2$^{\dagger\ddagger}$ | **85.2$^{\dagger\ddagger}$** | 75.9$^{\dagger\ddagger}$ | 79.2$^{\dagger\ddagger}$ |
| Correct > Irrelevant | $\Delta\Omega^{(\text{prompt w/ A})}$ | 83.2 | 39.7 | 52.2 | **88.5** | 65.9 |
| | $\Delta\Omega^{(\text{prompt w/o A})}$ | **93.6** | 68.4 | 79.9 | 72.4 | 78.6 |
| | $\Delta\Psi^{(\text{log likelihood})}$ | 76.7 | 74.7$^{\dagger\ddagger}$ | 74.5$^{\dagger}$ | 80.6$^{\ddagger}$ | 76.6$^{\dagger}$ |
| | $\Delta\Psi^{(\text{model internals})}$ | 86.3$^{\dagger}$ | **90.5$^{\dagger\ddagger}$** | **89.6$^{\dagger\ddagger}$** | 70.6 | **84.3$^{\dagger\ddagger}$** |

Table 3: Proportion of examples where $\Delta$LL successfully distinguishes context quality using prompting-based and internal-based confidence estimators on TriviaQA. We compute $\Delta\text{LL}(Q, C_{\text{correct}}, C_{\text{incorrect}})$ and $\Delta\text{LL}(Q, C_{\text{correct}}, C_{\text{irrelevant}})$ separately for each example and report the fraction for which the result is greater than zero. $^{\dagger}$ and $^{\ddagger}$ indicate statistical significance compared to Prompt w/ A and Prompt w/o A, respectively (p-value $< 0.05$, two-sided permutation test).

In contrast, the classifier trained on features extracted with the *Logit Lens* show strong performance, yielding the highest AUC-ROC scores on both datasets for most models. *Tuned Lens* performs comparably to *Logit Lens*, despite being developed to improve upon *Logit Lens* by better aligning intermediate hidden states with the output distribution through affine transformations. We hypothesize that the only distinction between the Tuned Lens and the Logit Lens is the affine transformation, and thus the corresponding scaling and shifting of activations do not contribute additional predictive information.

Surprisingly, classifiers trained directly on hidden states perform nearly as well as those based on Logit Lens features on both datasets. This finding indicates that the vanilla activations already encode information about the correctness of the model's output. In Figure 2, we observe that performance improves with depth, suggesting that later layers refine and consolidate decision-relevant signals. This result implies that intermediate decoding methods may not be necessary to predict correctness and the strong performance of classifiers trained on first token hidden states across models may allow for correctness auditing of model responses early in the generation process.

Lastly, we find that PKS alone is strongly predictive of correctness on TriviaQA, despite being explicitly designed to just measure the influence of feedforward networks on token representations. This indicates that the feedforward layers of an LLM impart information onto the activations differently based on how in-distribution a given query is. However, PKS performs close to random chance for MMLU, suggesting that its discriminatory power may be limited to open-ended question answering and not extend to single-token multiple choice questions.

## 6.2 RQ2: Can We Estimate Context Efficacy Directly from Model Internals?

Table 3 shows that internals-based confidence estimation significantly outperforms prompting-based methods when distinguishing between correct and incorrect contexts. This result is striking: while models fail to express higher confidence when generating answers conditioned on correct context, their internal activations nonetheless reflect this difference. All models exceed the 50% random baseline by a wide margin. We do not include results for LLaMA 2 13B due to computational constraints, nor for Gemma 2 9B due to the inapplicability of ECS to its architecture. Further explanation is provided in §F.1.

When comparing correct and irrelevant contexts, *Prompting without Answer* performs well, often on par with internal-based estimation and consistently outperforms the *Prompting with Answer* baseline. This suggests that including the model's own answer in the prompt may mislead it, especially when the answer is incorrect due to misleading context. Without the generated answer, however, models appear better able to distinguish irrelevant contexts. This is paradoxical: despite being able to differentiate between relevant and irrelevant contexts, the underlying LLM remains strongly influenced by irrelevant context (see Table 1), suggesting that recognition alone is insufficient to steer the model towards an accurate answer. We suspect that during training, the model is rarely given irrelevant context and learns to implicitly trust contextual information.

# 7 DISCUSSION

## 7.1 RQ1

**Prompting Baselines**  Here, we look at how well calibrated the prompting-based approaches are by looking at a reliability diagram and measuring smooth expected calibration error (Smooth ECE) (Błasiok & Nakkiran, 2023). In our reliability diagrams, we plot the confidence (x-axis) vs. accuracy (y-axis) and plot the line $y = x$ indicating a perfectly calibrated system. Figures 3–6 in our Appendix show that all prompting baselines for all LLMs for both TriviaQA and MMLU are systematically overconfident (*i.e.*, predictions with $x\%$ confidence yield $< x\%$ accuracy).

**Logit Lens vs. Tuned Lens**  In addition to the comparison presented in Table 2, we further train separate classifiers on logit lens and tuned lens features of each individual layer to assess performance across layers on TriviaQA (see Figure 7 in Appendix). The performance of the two methods varies across layers without a consistent trend and neither consistently outperforms the other. We observe that earlier layers are also highly predictive of generation correctness, achieving performance comparable to later layers. This strengthens our finding that information about output correctness is available early on in the models' activations, underscoring the potential of early auditing.

**Logit/Tuned Lens: Feature Importance**  We analyze feature importance scores from random forest classifiers for TriviaQA in Figure 8 in the Appendix. Entropy and cross-entropy from the final layer emerge as the most important features across all models. This is intuitive: output token logits and distributions directly capture the model's confidence when generating the correct answer. Additionally, features from the last $\sim 5$ layers generally exhibit higher importance.

**Logit/Tuned Lens: AUC-ROC Curves of Features from External vs. Internal Layers**  To examine whether features from non-final layers (denoted as *Internal*) are as predictive as those from the final layer alone (denoted as *External*), we train separate classifiers using each feature set and compare their AUC-ROC curves, as shown in Figure 9 for TriviaQA and Figure 10 for MMLU. Interestingly, across all models, classifiers trained on internal layers achieve higher AUC scores than those using only the final layer. This suggests that intermediate representations can carry more information than the final layer alone for predicting generation correctness. See §E for other analyses.

## 7.2 RQ2

**Example Analysis**  We visualize ECS and PKS values across model layers and output tokens for selected TriviaQA examples in Figures 12, 13, and 14 in the Appendix. While PKS scores show only small variations across tokens and layers, later layers show noticeably higher values. Here, initial output tokens have higher PKS values than later ones. These patterns suggest that parametric knowledge accumulated in later layers has a stronger influence on the output distribution, particularly at the beginning of generation.

In contrast, ECS scores vary more across tokens, reflecting the model's selective use of context depending on the importance and influence of the context on each output token. Uninformative tokens tend to exhibit lower ECS scores. Across layers, ECS shows relatively little variation, indicating that attention to external context is distributed broadly across the network rather than being isolated to specific layers. While features continue to evolve across layers, the degree to which tokens attend to one another's features appears relatively stable.

# 8 CONCLUSION

Our experiments demonstrate that we can indeed predict the correctness of a model generation from model internals alone. In fact, with just the activations of first output token, we can predict correctness with about 75% accuracy, hinting that early auditing could be possible. Prompting, on the other hand, is poorly calibrated and thus has little utility. Additionally, using model internals based *contextualized log-likelihood*, we can estimate the efficacy of external context along two axis: correctness and relevancy. Taken together, our results suggest that deciphering model internals could provide valuable insight into making language models more trustworthy.

## REPRODUCIBILITY STATEMENT

We provide the dataset details, training details, model specifications, and prompts used in the main text or the Appendix to ensure our experiments are reproducible. We use mainstream and commonly-used software libraries, datasets, and models for our experiments. We publicly release our code.

## ETHICS STATEMENT

Large language models carry significant potential for misuse, both intentional and accidental. Our work aims to advance understanding of the decision-making processes in LLMs and to identify hallucinations in model generations, thereby helping to mitigate some of the harms associated with the spread of inaccurate, model-generated content. Nevertheless, our methods should not be seen as a substitute for careful usage and verification; all model outputs must be independently checked for accuracy, particularly in domains where correctness is critical.

We emphasize that all datasets and models used in this study are available under permissive licenses for research purposes. We rely on instruction-tuned models, which have already undergone safety-related training. At the same time, we acknowledge that probing or intervening in model internals may alter or undermine this safety tuning. Understanding and exposing hidden mechanisms inside LLMs can yield valuable scientific insight, but it also carries risks: such methods could, in principle, be adapted to bypass alignment safeguards or weaken safety behaviors.

We therefore caution that techniques for manipulating internal representations should be applied with care and with an awareness of their broader implications. Our intention in presenting this work is to promote transparency, accountability, and safer deployment of LLMs, not to provide tools that could be used to compromise alignment or safety constraints.

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

## A    LIMITATIONS

This study has the following limitations. Several of our analyses are performed either in aggregate across token positions or only at the first output token. It may be valuable to study variations across token positions, especially in settings with longer model outputs. Although the TriviaQA and MMLU datasets cover diverse domains, we restrict our analysis to short-answer factual question-answering for computational feasibility. It is valuable to study the generalizability of our results to different tasks and domains; we leave this for future work. Second, we only experiment with instruction-tuned models. Investigating how pre-trained models differ from our analysis may reveal how post-training processes affect models' confidence estimates. In addition, experiments with larger models ($> 32B$) may yield more generalizable results.

## B  METHODOLOGY DETAILS FOR RQ1

### B.1  PROMPTING

Below are the prompts used for RQ1 to elicit confidence estimates from the model. We explore two prompting formats: *Prompting without Answer* and *Prompting with Answer*.

**Prompting without Answer** In this setting, the model is asked to estimate its confidence in being able to answer the question correctly, without actually providing an answer. The prompt is as follows:

```
For the question below, output your confidence in your ability to generate
the correct answer as an integer between 0 and 100, where 0 means complete
uncertainty and 100 means complete certainty. Provide only the confidence
score, without answering the question or including any explanations or
additional text.

Question:
```
{Question}
```
```

**Prompting with Answer** This format uses a multi-turn chat interaction. First, the model generates an answer. Then, in a second turn, it is asked to provide a confidence score for the generated answer.

*Turn 1:*

```
Answer the following question with a single word or phrase. Do not provide
explanations or additional context:

{Question}
```

*Turn 2:*

```
Please output your confidence in the correctness of the answer as an
integer between 0 and 100, where 0 means complete uncertainty and 100
means complete certainty. Output only the confidence score with no
explanations or additional text.
```

### B.2  LOGIT LENS AND TUNED LENS

**Logit Lens:** *Logit Lens* projects the intermediate hidden states directly into the vocabulary space via the unembedding matrix or final linear layer of the model (nostalgebraist, 2020). Concretely, given a hidden state $\mathbf{h}_\ell \in \mathbb{R}^d$ at layer $\ell$, and the unembedding matrix $\mathbf{W}_U \in \mathbb{R}^{V \times d}$ (where $V$ is the vocabulary size and $d$ is the hidden size), the layer-wise logits under the logit lens are computed as: $\mathbf{z}_\ell^{(\text{LL})} = \mathbf{W}_U \mathbf{h}_\ell$. For decoding, these logits are passed through a softmax as normal to produce a probability distribution over the vocabulary: $\mathbf{p}_\ell^{(\text{LL})} = \text{softmax}(\mathbf{z}_\ell^{(\text{LL})})$.

**Tuned Lens:** *Tuned Lens* first learns an affine transformation or *translator* for each layer and then projects the transformed hidden state directly to the vocabulary space via the unembedding matrix (Belrose et al., 2023). Since pretrained language models are typically not trained to project from intermediate hidden states to the unembedding matrix, there is often a mismatch. *Tuned Lens* remediates this by post-hoc training these *translators* at each layer, improving intermediate hidden states alignment in both magnitude and direction to the unembedding matrix. For each layer $\ell$, the learned affine map $(\mathbf{A}_\ell, \mathbf{b}_\ell)$ transforms the hidden state $\mathbf{h}_\ell$ before multiplying with the unembedding matrix: $\mathbf{z}_\ell^{(\text{TL})} = \mathbf{W}_U(\mathbf{A}_\ell \mathbf{h}_\ell + \mathbf{b}_\ell)$. The resulting logits can be passed through a softmax producing a probability distribution over the vocabulary: $\mathbf{p}_\ell^{(\text{TL})} = \text{softmax}(\mathbf{z}_\ell^{(\text{TL})})$. The *translators* are trained to minimize the KL divergence between $\mathbf{p}_\ell^{(\text{TL})}$ and the model's final output distribution on general-purpose pretraining data.

## B.3 Input Feature Details for Logit Lens and Tuned Lens

**Input Feature 1: Entropy**   Entropy measures the uncertainty in a probability distribution. For us, entropy estimates how peaky (or uniform) the distribution over the vocabulary is at each token position at each layer. Given a distribution over the vocabulary using either *Logit Lens* (LL) or *Tuned Lens* (TL), $\mathbf{p}_\ell^{(\mathrm{LL—TL})} = \{p_\ell^{(\mathrm{LL—TL})}(i)\}_{i=1}^V$ at layer $\ell$, the entropy $H_\ell$ is defined as:

$$H_\ell = -\sum_{i=1}^{V} p_\ell^{(\mathrm{LL—TL})}(i) \log p_\ell^{(\mathrm{LL—TL})}(i) \tag{6}$$

Note: these estimates are computed independently for each layer and sequence position.

**Input Feature 2: Rank of Output Token**   A probability of 0.01 in certain token positions may reflect the highest rank token if the preceding context is ambiguous (e.g. The ¡next_word¿), whereas could reflect a much lower rank token if the context is less ambiguous (e.g. Harriet Tub¡next_word¿) The rank of the output next token at intermediate layers loosely correlates with the probability, but can offer a more direct signal towards what token a decoding algorithm may likely generate. At layer $\ell$, we compute either the *Logit Lens* (LL) or *Tuned Lens* (TL) distribution $\mathbf{p}_\ell^{(\mathrm{LL—TL})}$. Let $\phi_\ell(t) := \log p_\ell^{(\mathrm{LL—TL})}(t)$ denote the log-probability of token $t$ under this distribution. We sort the vocabulary tokens by descending log-probability:

$$\phi_\ell\big(\sigma(1)\big) \geq \phi_\ell\big(\sigma(2)\big) \geq \cdots \geq \phi_\ell\big(\sigma(V)\big) \tag{7}$$

Then, the *rank* $r_\ell(y)$ of the target token $y$ is the index $k$ such that $\sigma(k) = y$. A rank of 1 indicates the model considers $y$ the most likely token at that layer. This is computed per layer and position.

**Input Feature 3: Top-$p$ Presence**   Nucleus-sampling is one of the most common decoding algorithms for language models (Holtzman et al., 2019), where rather than considering the entire distribution over the vocabulary, one considers only the highest probability tokens until a top-$p$ cumulative probability. This eliminates the chances of randomly sampling a very low probability token in the long-tail. Motivated by this, we compute a binary indicator of whether the target token appears within the top-$p$ nucleus set at each layer. Let $\mathbf{p}_\ell^{(\mathrm{LL—TL})}$ be sorted in descending order as $p_{(1)} \geq p_{(2)} \geq \cdots$. The top-$p$ nucleus set $V_p$ is the smallest set such that:

$$\sum_{i=1}^{k} p_{(i)} \geq p \tag{8}$$

The *top-$p$ presence* indicator $I_{p,\ell}(y)$ is defined as:

$$I_{p,\ell}(y) = \begin{cases} 1 & \text{if } y \in V_p \\ 0 & \text{otherwise} \end{cases} \tag{9}$$

**Input Feature 4: Cross-Entropy**   Cross-entropy quantifies how well the intermediate predictions match the output next token. At each layer $\ell$, it is computed using the log probability assigned to the target token $y$ under the logit lens or tuned lens distribution:

$$\mathrm{CE}_\ell(y) = -\log p_\ell^{(\mathrm{LL—TL})}(y) \tag{10}$$

This value is lower when the model assigns higher confidence to the output token. It is evaluated per layer and token position.

## C   Methodology Details for RQ2

### C.1   Prompting

Below are the prompts used for RQ2 to elicit confidence estimates from the model. We explore two prompting formats: *Prompting without Answer* and *Prompting with Answer*.

**Prompting without Answer** In this setting, the model is provided with context and is asked to estimate its confidence in answering the question correctly, without actually generating an answer. The prompt used is as follows:

```
For the question below, output your confidence in your ability to generate
the correct answer as an integer between 0 and 100, where 0 means complete
uncertainty and 100 means complete certainty. Provide only the confidence
score, without answering the question or including any explanations or
additional text.

Context:
```
{context}
```

Question:
```
{question}
```
```

**Prompting with Answer** This format uses a multi-turn chat interaction. In the first turn, the model is provided with context and asked to generate an answer to the question. In the second turn, it is asked to estimate its confidence in the answer it just provided.

*Turn 1:*

```
Using the following context, answer the question that follows with a
single word or phrase. Do not provide explanations or additional context:

> Context:
{Context}

> Question:
{Question}
```

*Turn 2:*

```
Please output your confidence in the correctness of the answer as an
integer between 0 and 100, where 0 means complete uncertainty and 100
means complete certainty. Output only the confidence score with no
explanations or additional text.
```

## C.2 REASONING FOR CONTEXT COMPARISON

**Correct vs. Incorrect Context** A correct and relevant context ($C_{\text{correct}}$) directly supports the correct answer with accurate factual information. This should increase the model's confidence, especially when its parametric knowledge alone is insufficient. In contrast, an incorrect but relevant context ($C_{\text{incorrect}}$) may appear plausible but contains factual errors. While it may influence the model's output, it should not increase confidence as much—especially if the model can detect inconsistencies or is sensitive to contradiction. Thus, we expect higher confidence in the presence of $C_{\text{correct}}$ than $C_{\text{incorrect}}$.

**Correct vs. Irrelevant Context** An irrelevant context ($C_{\text{irrelevant}}$) is topically unrelated to the question and provides no useful information for answering it. Ideally, the model should recognize its lack of utility and ignore it, leading to little or no increase in confidence compared to the no-context baseline. In contrast, $C_{\text{correct}}$ provides directly useful information, so the model should become more confident in its answer. Therefore, the confidence gain should be higher with $C_{\text{correct}}$ than with $C_{\text{irrelevant}}$.

# D EXPERIMENT SETUP DETAILS

## D.1 DATASET DETAILS

| Question | Which island in Kent is the second largest of England's isles? |
|---|---|
| Correct Answers | "Shurland Hall", "Isle of Sheppy", "Shurland House", "Isle of Sheppey" |
| Question | Rita Coolidge sang the title song for which Bond film? |
| Correct Answers | "Kamal kahn", "List of Bond girls in Octopussy", "Magda (James Bond)", "List of James Bond allies in Octopussy", "Vijay (James Bond)", "Bond 13", "Octopussy (character)", "Penelope Smallbone", "Octopussy", "General Orlov", "Kamal Khan", "Octopussy (film)", "List of James Bond villains in Octopussy", "Jim Fanning (James Bond)" |
| Question | What was invented by Jonas Hanway in the late 1750s? |
| Correct Answers | "Umbrella", "Umbrela", "History of the umbrella", "Unbrella", "Beach umbrella", "Parasols", "Windproof umbrella", "Beach parasol", "Parasol", "History of the Umbrella", "Umbrellas" |

Table 4: Examples from the TriviaQA dataset.

| Question | Which of the following bones develop by endochondral ossification? |
|---|---|
| Choices | A: "The ribs", B: "The ribs and sternum", C: "The ribs, sternum and clavicle", D: "The ribs, sternum, clavicle and vertebrae" |
| Correct Answer | B |
| Question | Which of the following gives the total spin quantum number of the electrons in the ground state of neutral nitrogen (Z = 7)? |
| Choices | A: "1/2", B: "1", C: "3/2", D: "5/2" |
| Correct Answer | C |
| Question | The Hawthorn Studies are most associated with which writer? |
| Choices | A: "Mary Parker Follett", B: "Elton Mayo", C: "Lillian Gilbreth", D: "Frederick Taylor" |
| Correct Answer | B |

Table 5: Examples from the MMLU dataset.

**Quality Filtering:** TriviaQA includes multiple retrieved Wikipedia or web documents as evidence for each example, which we repurpose to serve as (correct) context for RQ2 experiments. While the documents are generally of high quality, manual inspection reveals a small percentage of examples where the documents do not contain enough information to answer the question. As a result, we exclude any example where no retrieved document contains one of the ground-truth answers and at least 60% of entities extracted from the question.[3] This yields a final set of 6,557 examples, which we use for all our experiments. No quality filtering is necessary for MMLU. Table 4 and Table 5 present example questions along with their corresponding ground-truth answers from TriviaQA and MMLU.

**External Context Construction:** Concatenating multiple long documents and analyzing these is challenging for interpretability methods that normally operate at the single-token or very few token level. To address this, we select a single evidence document from TriviaQA that meets the quality filtering criteria described above. If the selected document exceeds 500 tokens, we use GPT-4o to summarize it down to 500 tokens, using the LLaMA-3-8B tokenizer for consistency. This summary is termed the "correct" context for the experiments for RQ2. To generate "incorrect" contexts, we take the "correct" context for a given question $Q$ and prompt GPT-4o to replace all references to the ground-truth answer with plausible but incorrect alternatives. Crucially, we keep the rest of the text unchanged, which makes the surrounding context relevant to $Q$. We also obtain "irrelevant" contexts by sampling "correct" contexts from a different question such that the context has no lexical overlap with the ground-truth answer. Additionally, we require that the SentenceBERT (Reimers & Gurevych,

---

[3]We use the FLAIR Sequence Tagger for entity extraction (Akbik et al., 2019).

2019) embeddings using `all-MiniLM-L6-v2` of the candidate context have a cosine similarity $< 0.3$ with the "correct" context for that question $Q$. This context construction process is grounded in the original "correc" context, and the LLM is used solely for straightforward tasks such as summarization and term replacement.

**Prompting Formats:** Below are the prompt formats used to generate answers from models on TriviaQA questions, either with or without supporting context.

*Without context:*

```
Answer the following question with a single word or phrase. Do not provide
explanations or additional context:

{Question}
```

*With context:*

```
Using the following context, answer the question that follows with a
single word or phrase. Do not provide explanations or additional context:

> Context:
{Context}

> Question:
{Question}
```

We use the following prompt format to generate answers from models for MMLU questions.

```
The following is a multiple-choice question. Please choose the most
suitable one among A, B, C and D as the answer to this question. Only
output the choice identifier (A, B, C, or D) and nothing else. Do not
provide explanations or additional context.

{Question}
```

**Evaluation:** While the TriviaQA evaluation framework relies on exact match after text normalization between the candidate answer and one of the ground-truth answers, this rule-based approach can result in significant false negatives when the answer is correct but not phrased in exactly the same wording as the ground-truth answers. We use GPT-4o to assess correctness when exact match is not found due to its performance in factual QA evaluation (Zheng et al., 2023; Gu et al., 2024). Specifically, if the generated answer does not exactly match any of the ground-truth answers, we prompt GPT-4o with the question $Q$, the ground-truth answers $A*$, and the generated answer $A$ and ask it to judge correctness. Unattempted questions are marked incorrect. This improves robustness, especially because LLMs can phrase answers in slightly different but equivalent ways and may not exactly follow a given output format, especially when minimally post-trained. Since MMLU questions are multiple-choice, we ask the model to only output the choice identifier and rely on regex-based answer extraction and verification.

### D.2 RQ1 SETUP DETAILS

**Logit-Lens and Tuned-Lens** We include all top-$p$ values from the set $\{0.5, 0.9, 0.95, 0.99\}$ as input features when training our classifier using Logit-Lens and Tuned-Lens representations. We extract the statistics described in §B.3 (entropy, rank of the correct token, top-$p$ presence, and cross-entropy loss) from the decoded logits for each model layer. We then train our classifier $f$ using these features to predict whether a generated answer is correct or not.

**Hidden States** We train a separate classifier on the hidden states extracted from each transformer layer during the generation of the first answer token. We focus on the first token because, due to the

autoregressive nature of transformer models, the output at this initial step is conditioned solely on the input prompt and is unaffected by previously generated tokens (Zhao et al., 2024). For each example, per-dimension mean and standard deviation values used for calculating z-scores are obtained by collecting the first-output-token hidden states from each layer for questions from an unused split of the TriviaQA and MMLU datasets. This avoids any sharing of information across examples.

**Classifier:** In practice, one could use any classifier here. Subramani et al. (2025) show that random forests are best able to recalibrate tool-using agents when using model internals as features in comparison to other simple classifiers such as logistic regression. In our experiments, we use a random forest due to its efficacy, simplicity, and interpretability. We use grid search and five-fold cross-validation over a small subset (1024 examples) of our training set to set hyperparameters for the random forest classifier.

During hyperparameter tuning, a grid search was conducted over the following parameters for the Random Forest classifier: the number of estimators ($\texttt{n\_estimators} \in \{100, 200, 300\}$), maximum tree depth ($\texttt{max\_depth} \in \{\text{None}, 10, 20, 30\}$), minimum number of samples required to split an internal node ($\texttt{min\_samples\_split} \in \{2, 5\}$), minimum number of samples required at a leaf node ($\texttt{min\_samples\_leaf} \in \{1, 2\}$), the number of features considered for splitting at each node ($\texttt{max\_features} \in \{\text{"sqrt"}, \text{"log2"}, \text{None}\}$), and class weighting strategies ($\texttt{class\_weight} \in \{\text{"balanced"}, \text{"balanced\_subsample"}, \text{None}\}$). The hyperparameters were set to the following for random forest classifiers trained on Logit Lens, Tuned Lens, and PKS features: $\texttt{n\_estimators} = 300$, $\texttt{max\_depth} = \text{None}$, $\texttt{max\_features} = \text{"log2"}$, $\texttt{min\_samples\_leaf} = 1$, $\texttt{min\_samples\_split} = 2$, $\texttt{class\_weight} = \text{"balanced"}$. The hyperparameters were set to the following for random forest classifiers trained on hidden state features: $\texttt{n\_estimators} = 300$, $\texttt{max\_depth} = 10$, $\texttt{max\_features} = \text{"sqrt"}$, $\texttt{min\_samples\_leaf} = 1$, $\texttt{min\_samples\_split} = 2$, $\texttt{class\_weight} = \text{"balanced\_subsample"}$.

# E    ADDITIONAL RESULTS FOR RQ1

## E.1    PROMPTING

For prompting, we evaluate all integer threshold values from 0 to 100 and select the one that yields the highest accuracy. Accordingly, the performance reported for prompting methods reflects the best-performing threshold. Figure 3 and Figure 5 present the Smooth ECE scores of various models using prompting with answers on TriviaQA and MMLU respectively, while Figure 4 and Figure 6 show the scores for prompting without answers.

## E.2    LOGIT LENS

**Logit Lens VS. Tuned Lens**    Figure 7 shows the layerwise AUC-ROC performance of the Logit Lens and Tuned Lens on LLaMA 3 8B, LLaMA 2 13B, and LLaMA 2 7B. We do not observe a consistent trend in the relative performance that would clearly indicate which method is superior. However, both the Logit Lens and Tuned Lens exhibit predictive capability from very early layers.

**Feature Importance**    We present the Logit Lens feature importance heatmaps for all six models in Figure 8 based on the trained classifiers. Top-$k$ features are omitted due to their consistently low importance across all layers and models.

**ROC Curve: External VS. Internal**    Figure 9 shows the ROC curve comparisons between classifiers trained solely on last-layer features ("external") and those trained on internal-layer features ("internal") across all six models. The internal classifier achieves performance comparable to using features from all layers, while consistently outperforming the external classifier across all tested models.

## E.3    HIDDEN STATES

**Hidden States: Effect of Z-Score Normalization**    To evaluate the impact of z-score normalization on classifier performancewe obtain means and variances per hidden state dimension for z-score

|                   | LLaMA 3 8B | LLaMA 2 13B | LLaMA 2 7B | Gemma 2 9B | Qwen 2.5 7B | Qwen 2.5 3B |
|-------------------|------------|-------------|------------|------------|-------------|-------------|
| Averaged ACC      | 0.778      | 0.710       | 0.729      | 0.770      | 0.737       | 0.734       |
| Averaged AUC ROC  | 0.774      | 0.772       | 0.804      | 0.741      | 0.790       | 0.803       |
| First token ACC   | 0.782      | 0.779       | 0.776      | 0.774      | 0.751       | 0.729       |
| First token AUC ROC | 0.790    | 0.847       | 0.835      | 0.747      | 0.826       | 0.812       |

Table 6: TriviaQA results for LogitLens comparing averaged vs. first token features.

|                   | LLaMA 3 8B | LLaMA 2 13B | LLaMA 2 7B |
|-------------------|------------|-------------|------------|
| Averaged ACC      | 0.780      | 0.703       | 0.736      |
| Averaged AUC ROC  | 0.777      | 0.768       | 0.808      |
| First token ACC   | 0.779      | 0.775       | 0.775      |
| First token AUC ROC | 0.782    | 0.846       | 0.829      |

Table 7: TriviaQA results for TunedLens comparing averaged vs. first token features.

normalization on auxiliary subsets of TriviaQA and MMLU to ensure the distribution remains similar to the train and test examples while avoiding leakage of information across examples. To evaluate the impact of z-score normalization on classifier performance, we compare models trained on hidden state features with and without normalization and find virtually identical performance (see Figure 11). This suggests that the classifier is relatively insensitive to the absolute magnitude of hidden state feature values across different dimensions and examples.

### E.4 FIRST TOKEN FEATURES VS. AVERAGING ACROSS ALL OUTPUT TOKENS

We use first token features for all classifier training in RQ1. We study whether interpretability features are computed using only the first generated token or averaged across all generated tokens. First token features provide the practical benefit of enabling early auditing before the model produces a full response. At the same time, aggregating features across all tokens may capture additional signal present throughout the generation process. To evaluate these two approaches, we conduct an ablation across multiple models and multiple internal feature extraction methods, including LogitLens, TunedLens, hidden states, and PKS.

Across most models and datasets, averaging features over all generated tokens performs similarly or slightly better than using only the first token. The differences are most pronounced for LogitLens and TunedLens on MMLU, where full token averaging provides consistent improvements. On TriviaQA, both approaches achieve comparable performance with small method dependent variations. These findings suggest that while full token averaging can offer moderate gains, first token features remain competitive and are particularly useful when early auditing is desired for efficiency.

**TriviaQA Results** TriviaQA results for all methods are in Tables 6–9.

**MMLU Results** MMLU results are shown in Tables 10–12. Here, averaging across tokens provides more consistent benefits, particularly for LogitLens and TunedLens.

### E.5 MLP CLASSIFIER VS. RANDOM FOREST CLASSIFIER

This section investigates whether more expressive classifiers can further improve correctness prediction. Our main experiments use a random forest classifier due to its simplicity, robustness, and low tuning requirements. To assess whether a more flexible model can extract additional signal from interpretability features, we train a multi layer perceptron (MLP) classifier across all models and datasets. The MLP has two hidden layers with 256 and 64 units, uses ReLU activations, applies L2 regularization with coefficient 0.001, and is optimized with Adam using adaptive learning rates and early stopping.

The results reveal that the MLP achieves performance comparable to or better than the random forest classifier across most settings. Improvements are especially pronounced on MMLU, where the MLP consistently reaches higher accuracy and AUC ROC. These findings indicate that correctness

| | LLaMA 3 8B | LLaMA 2 13B | LLaMA 2 7B | Gemma 2 9B | Qwen 2.5 7B | Qwen 2.5 3B |
|---|---|---|---|---|---|---|
| Averaged ACC | 0.739 | 0.714 | 0.707 | 0.773 | 0.782 | 0.768 |
| Averaged AUC ROC | 0.611 | 0.670 | 0.633 | 0.584 | 0.769 | 0.754 |
| First token ACC | 0.759 | 0.679 | 0.702 | 0.785 | 0.782 | 0.749 |
| First token AUC ROC | 0.647 | 0.631 | 0.647 | 0.616 | 0.774 | 0.735 |

Table 8: TriviaQA results for Hidden States comparing averaged vs. first token features.

| | LLaMA 3 8B | LLaMA 2 7B | Gemma 2 9B | Qwen 2.5 7B | Qwen 2.5 3B |
|---|---|---|---|---|---|
| Averaged ACC | 0.733 | 0.709 | 0.743 | 0.650 | 0.695 |
| Averaged AUC ROC | 0.729 | 0.743 | 0.723 | 0.715 | 0.752 |
| First token ACC | 0.737 | 0.668 | 0.733 | 0.659 | 0.684 |
| First token AUC ROC | 0.740 | 0.741 | 0.642 | 0.702 | 0.731 |

Table 9: TriviaQA results for PKS comparing averaged vs. first token features.

prediction can benefit from more expressive classifiers, although both approaches remain competitive depending on the method and dataset.

**TriviaQA Results**  Full TriviaQA results are shown in Table 13.

**MMLU Results**  MMLU results appear in Table 14. Compared to the random forest classifier, the MLP achieves large gains for Logit Lens and Tuned Lens and modest gains for Hidden States and PKS.

# F  ADDITIONAL RESULTS FOR RQ2

## F.1  NON-APPLICABILITY OF ECS ON GEMMA 2 9B

Gemma 2 9B employs sliding-window attention on every even-numbered layer, which makes it infeasible to implement ECS in a way that correctly tracks the top-k attention scores. Consequently, we exclude Gemma 2 9B from our analysis for RQ2.

## F.2  EXAMPLES WITH PKS AND ECS SCORES

We illustrate how PKS and ECS scores evolve as the model generates output tokens across layers, using several examples shown in Figures 12, 13, and 14.

# G  COMPUTATIONAL COSTS

All experiments were conducted using two A6000 GPUs, with a total compute time of under 1,000 hours.

| | LLaMA 3 8B | LLaMA 2 13B | LLaMA 2 7B | Gemma 2 9B | Qwen 2.5 7B | Qwen 2.5 3B |
|---|---|---|---|---|---|---|
| Averaged ACC | 0.836 | 0.717 | 0.684 | 0.814 | 0.822 | 0.803 |
| Averaged AUC ROC | 0.941 | 0.789 | 0.757 | 0.904 | 0.945 | 0.908 |
| First token ACC | 0.705 | 0.692 | 0.699 | 0.769 | 0.815 | 0.673 |
| First token AUC ROC | 0.798 | 0.771 | 0.767 | 0.843 | 0.939 | 0.711 |

Table 10: MMLU results for LogitLens comparing averaged vs. first token features.

| | LLaMA 3 8B | LLaMA 2 13B | LLaMA 2 7B |
|---|---|---|---|
| Averaged ACC | 0.826 | 0.728 | 0.680 |
| Averaged AUC ROC | 0.928 | 0.815 | 0.768 |
| First token ACC | 0.695 | 0.671 | 0.684 |
| First token AUC ROC | 0.770 | 0.752 | 0.760 |

Table 11: MMLU results for TunedLens comparing averaged vs. first token features.

| | LLaMA 3 8B | LLaMA 2 7B | Gemma 2 9B | Qwen 2.5 7B | Qwen 2.5 3B |
|---|---|---|---|---|---|
| Averaged ACC | 0.605 | 0.543 | 0.705 | 0.691 | 0.618 |
| Averaged AUC ROC | 0.537 | 0.543 | 0.555 | 0.541 | 0.538 |
| First token ACC | 0.588 | 0.529 | 0.708 | 0.694 | 0.615 |
| First token AUC ROC | 0.524 | 0.510 | 0.526 | 0.565 | 0.542 |

Table 12: MMLU results for PKS comparing averaged vs. first token features.

| Method | Source | Metric | LLaMA 3 8B | LLaMA 2 13B | LLaMA 2 7B | Gemma 2 9B | Qwen 2.5 7B | Qwen 2.5 3B |
|---|---|---|---|---|---|---|---|---|
| Logit Lens | Table 2 | ACC | 0.782 | 0.779 | 0.776 | 0.774 | 0.751 | 0.729 |
| Logit Lens | Table 2 | AUC ROC | 0.790 | 0.847 | 0.835 | 0.747 | 0.826 | 0.812 |
| Logit Lens | MLP | ACC | 0.788 | 0.778 | 0.760 | 0.770 | 0.711 | 0.686 |
| Logit Lens | MLP | AUC ROC | 0.790 | 0.844 | 0.820 | 0.758 | 0.764 | 0.755 |
| Tuned Lens | Table 2 | ACC | 0.779 | 0.775 | 0.775 | - | - | - |
| Tuned Lens | Table 2 | AUC ROC | 0.782 | 0.846 | 0.829 | - | - | - |
| Tuned Lens | MLP | ACC | 0.780 | 0.770 | 0.748 | - | - | - |
| Tuned Lens | MLP | AUC ROC | 0.784 | 0.827 | 0.808 | - | - | - |
| Hidden States | Table 2 | ACC | 0.739 | 0.714 | 0.707 | 0.773 | 0.782 | 0.768 |
| Hidden States | Table 2 | AUC ROC | 0.611 | 0.670 | 0.633 | 0.584 | 0.769 | 0.754 |
| Hidden States | MLP | ACC | 0.759 | 0.679 | 0.702 | 0.785 | 0.782 | 0.749 |
| Hidden States | MLP | AUC ROC | 0.647 | 0.631 | 0.647 | 0.616 | 0.774 | 0.735 |
| PKS | Table 2 | ACC | 0.733 | 0.725 | 0.709 | 0.743 | 0.650 | 0.695 |
| PKS | Table 2 | AUC ROC | 0.729 | 0.768 | 0.743 | 0.723 | 0.715 | 0.752 |
| PKS | MLP | ACC | 0.728 | 0.691 | 0.675 | 0.729 | 0.667 | 0.658 |
| PKS | MLP | AUC ROC | 0.715 | 0.744 | 0.722 | 0.595 | 0.731 | 0.730 |

Table 13: TriviaQA results comparing the random forest classifier (Table 2) with an MLP classifier across all methods and models.

| Method | Source | Metric | LLaMA 3 8B | LLaMA 2 13B | LLaMA 2 7B | Gemma 2 9B | Qwen 2.5 7B | Qwen 2.5 3B |
|---|---|---|---|---|---|---|---|---|
| Logit Lens | Table 2 | ACC | 0.705 | 0.692 | 0.699 | 0.769 | 0.815 | 0.673 |
| Logit Lens | Table 2 | AUC ROC | 0.798 | 0.771 | 0.767 | 0.843 | 0.939 | 0.711 |
| Logit Lens | MLP | ACC | 0.901 | 0.858 | 0.834 | 0.940 | 0.974 | 0.729 |
| Logit Lens | MLP | AUC ROC | 0.961 | 0.933 | 0.914 | 0.975 | 0.990 | 0.796 |
| Tuned Lens | Table 2 | ACC | 0.705 | 0.692 | 0.699 | - | - | - |
| Tuned Lens | Table 2 | AUC ROC | 0.798 | 0.771 | 0.767 | - | - | - |
| Tuned Lens | MLP | ACC | 0.901 | 0.858 | 0.834 | - | - | - |
| Tuned Lens | MLP | AUC ROC | 0.961 | 0.933 | 0.914 | - | - | - |
| Hidden States | Table 2 | ACC | 0.744 | 0.686 | 0.672 | 0.801 | 0.777 | 0.746 |
| Hidden States | Table 2 | AUC ROC | 0.740 | 0.684 | 0.576 | 0.736 | 0.728 | 0.733 |
| Hidden States | MLP | ACC | 0.748 | 0.679 | 0.664 | 0.779 | 0.770 | 0.730 |
| Hidden States | MLP | AUC ROC | 0.753 | 0.676 | 0.585 | 0.736 | 0.741 | 0.723 |
| PKS | Table 2 | ACC | 0.605 | - | 0.543 | 0.705 | 0.691 | 0.618 |
| PKS | Table 2 | AUC ROC | 0.537 | - | 0.543 | 0.555 | 0.541 | 0.538 |
| PKS | MLP | ACC | 0.596 | - | 0.530 | 0.709 | 0.694 | 0.617 |
| PKS | MLP | AUC ROC | 0.524 | - | 0.518 | 0.507 | 0.534 | 0.546 |

Table 14: MMLU results comparing the random forest classifier (Table 2) with an MLP classifier across all methods and models.

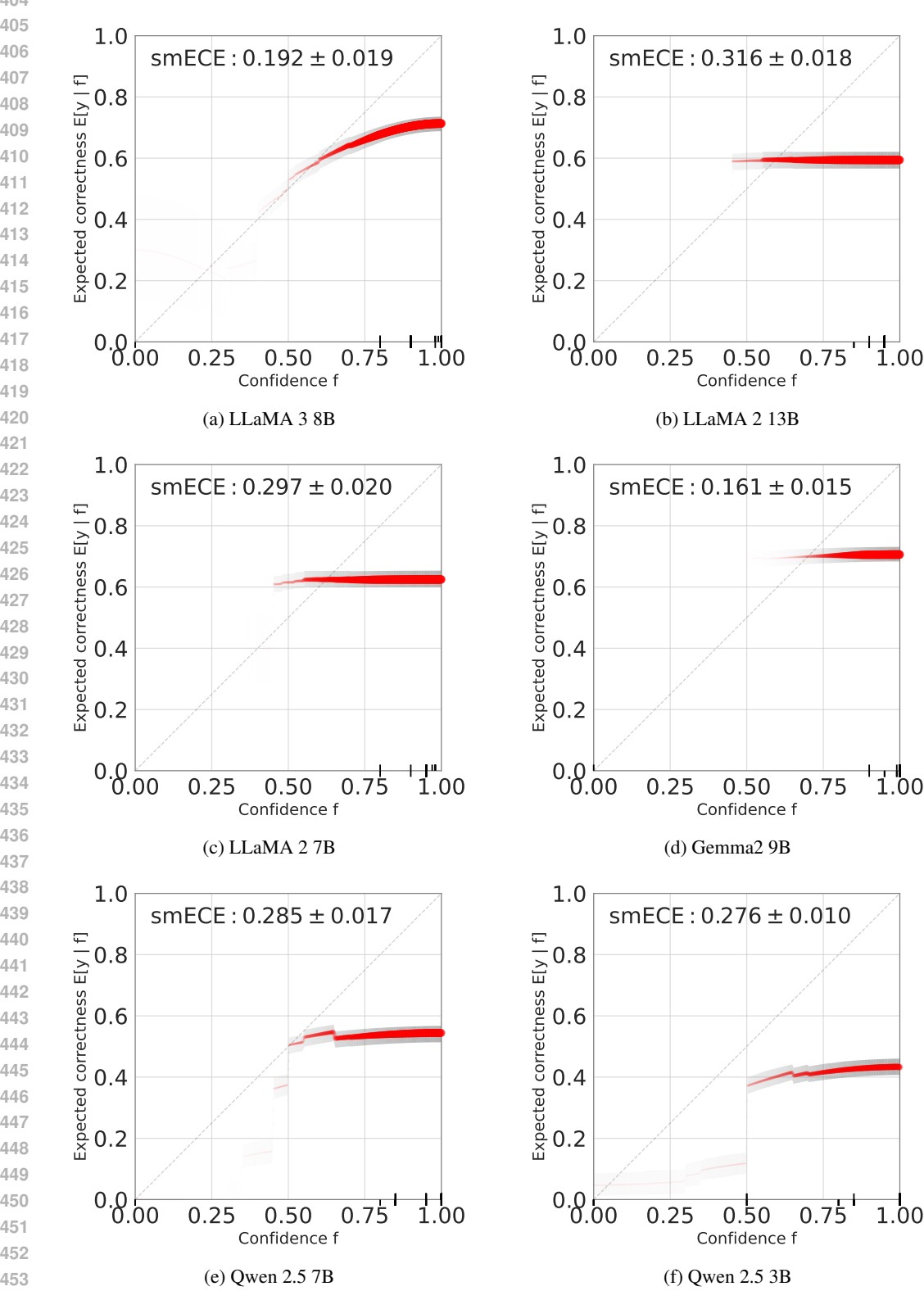

Figure 3: Smooth ECE scores for prompting with answer on TriviaQA. Predicted confidence $f$ is the model's stated probability of being correct, and $E[y|f]$ is the actual accuracy observed at that confidence level.

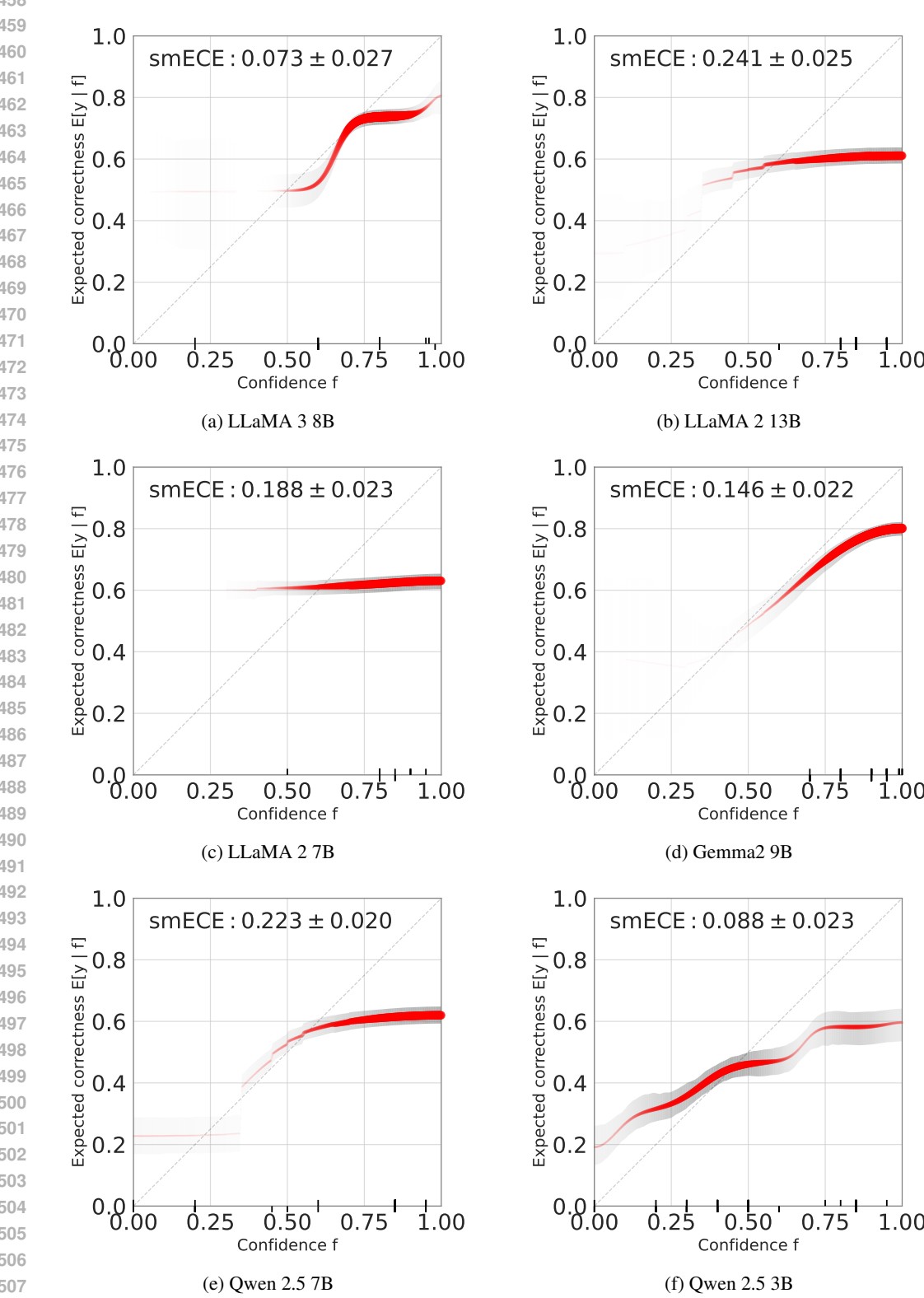

Figure 4: Smooth ECE scores for prompting without answer on TriviaQA. Predicted confidence $f$ is the model's stated probability of being correct, and $E[y|f]$ is the actual accuracy observed at that confidence level.

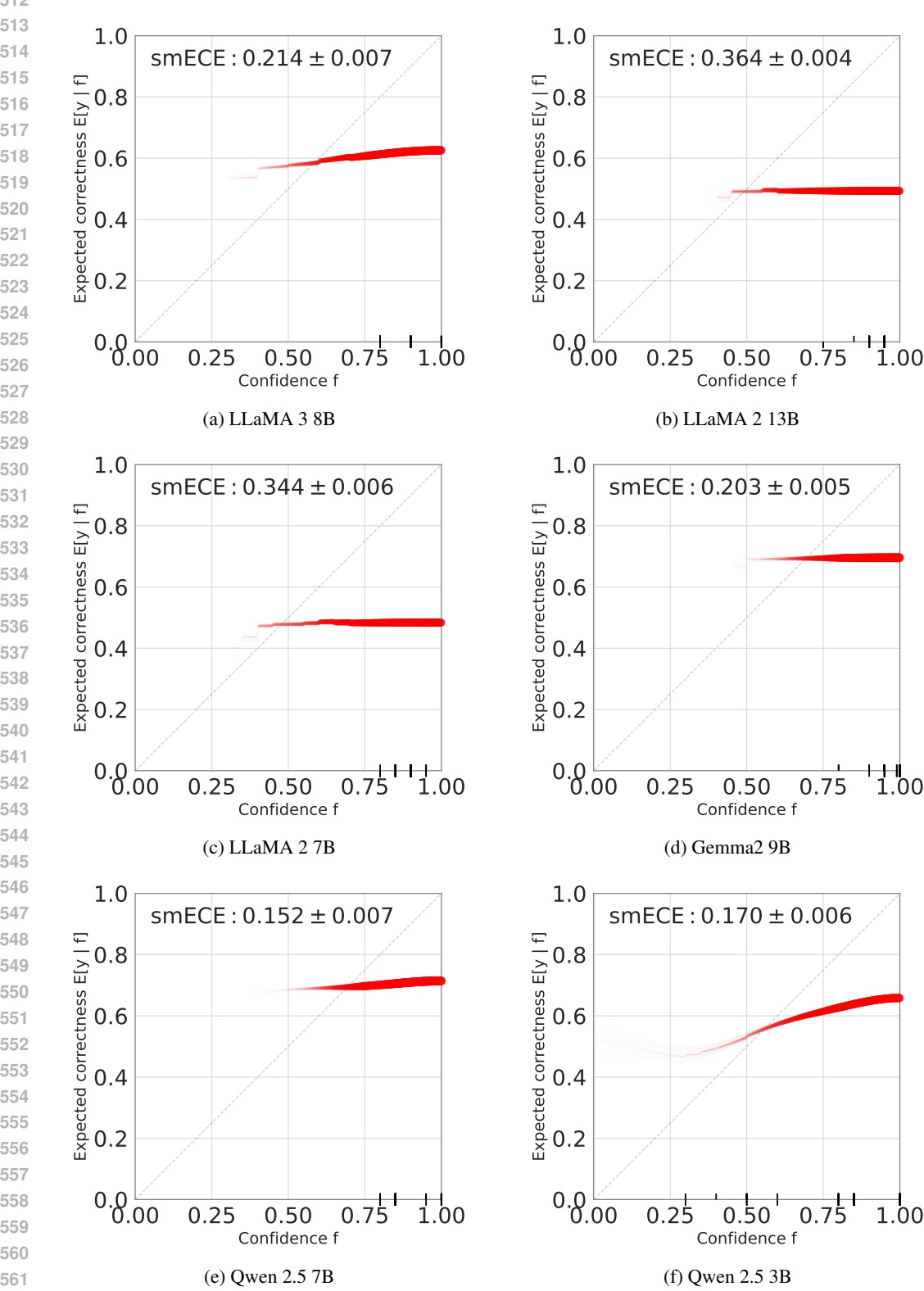

Figure 5: Smooth ECE scores for prompting with answer on MMLU. Predicted confidence $f$ is the model's stated probability of being correct, and $E[y|f]$ is the actual accuracy observed at that confidence level.

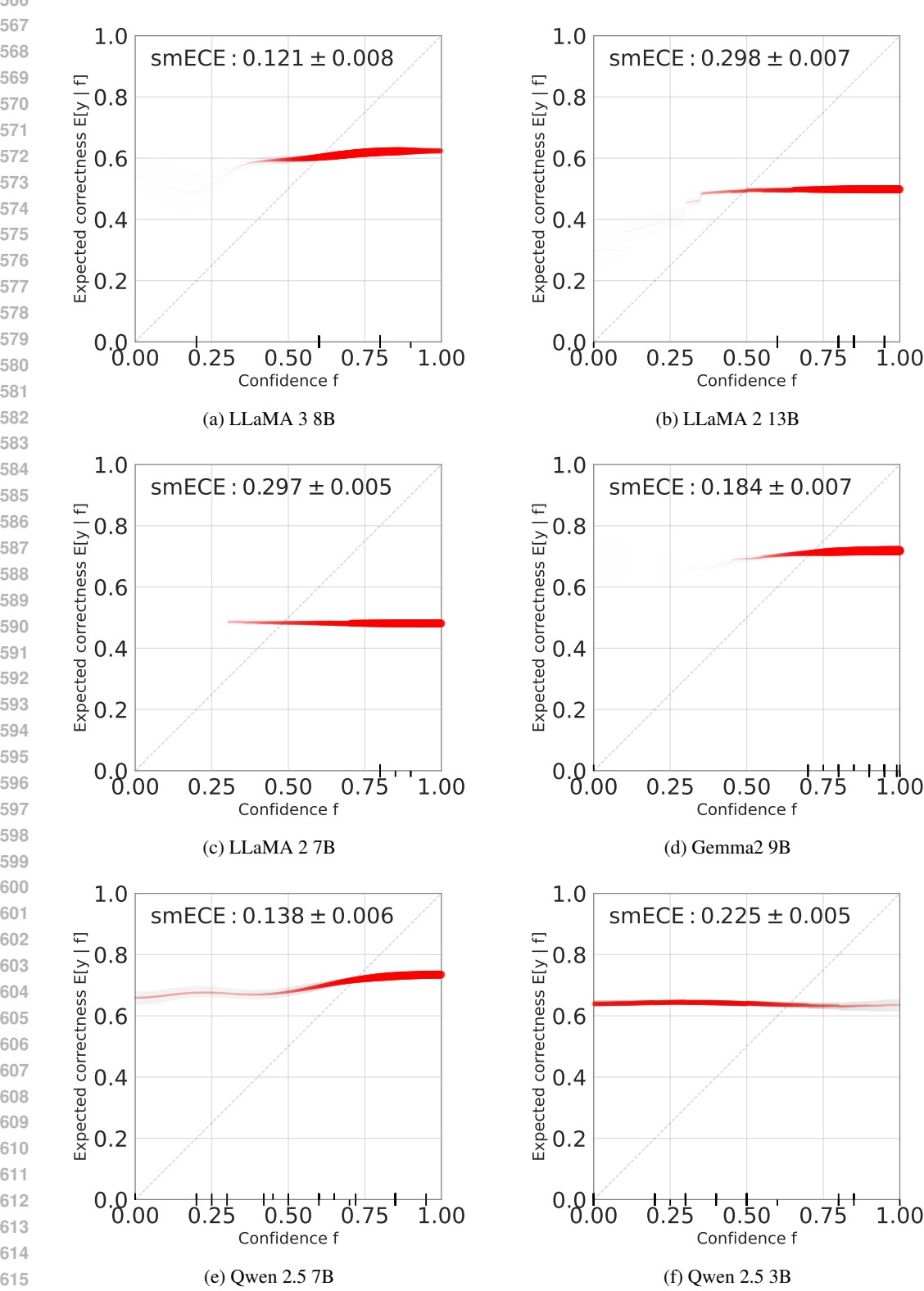

Figure 6: Smooth ECE scores for prompting without answer on MMLU. Predicted confidence $f$ is the model's stated probability of being correct, and $E[y|f]$ is the actual accuracy observed at that confidence level.

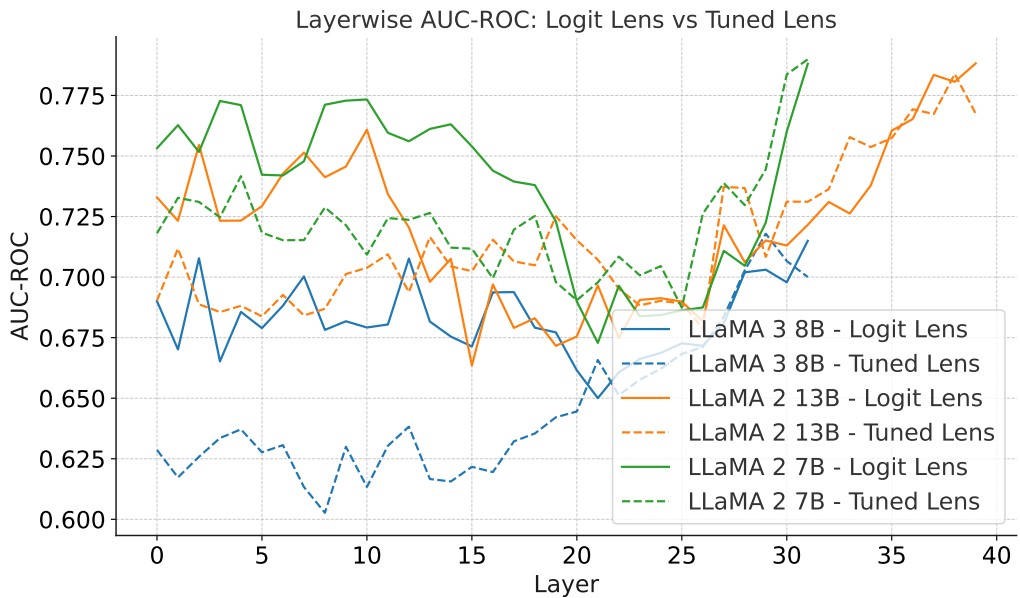

Figure 7: Layerwise AUC-ROC for Logit Lens (solid lines) and Tuned Lens (dashed lines).

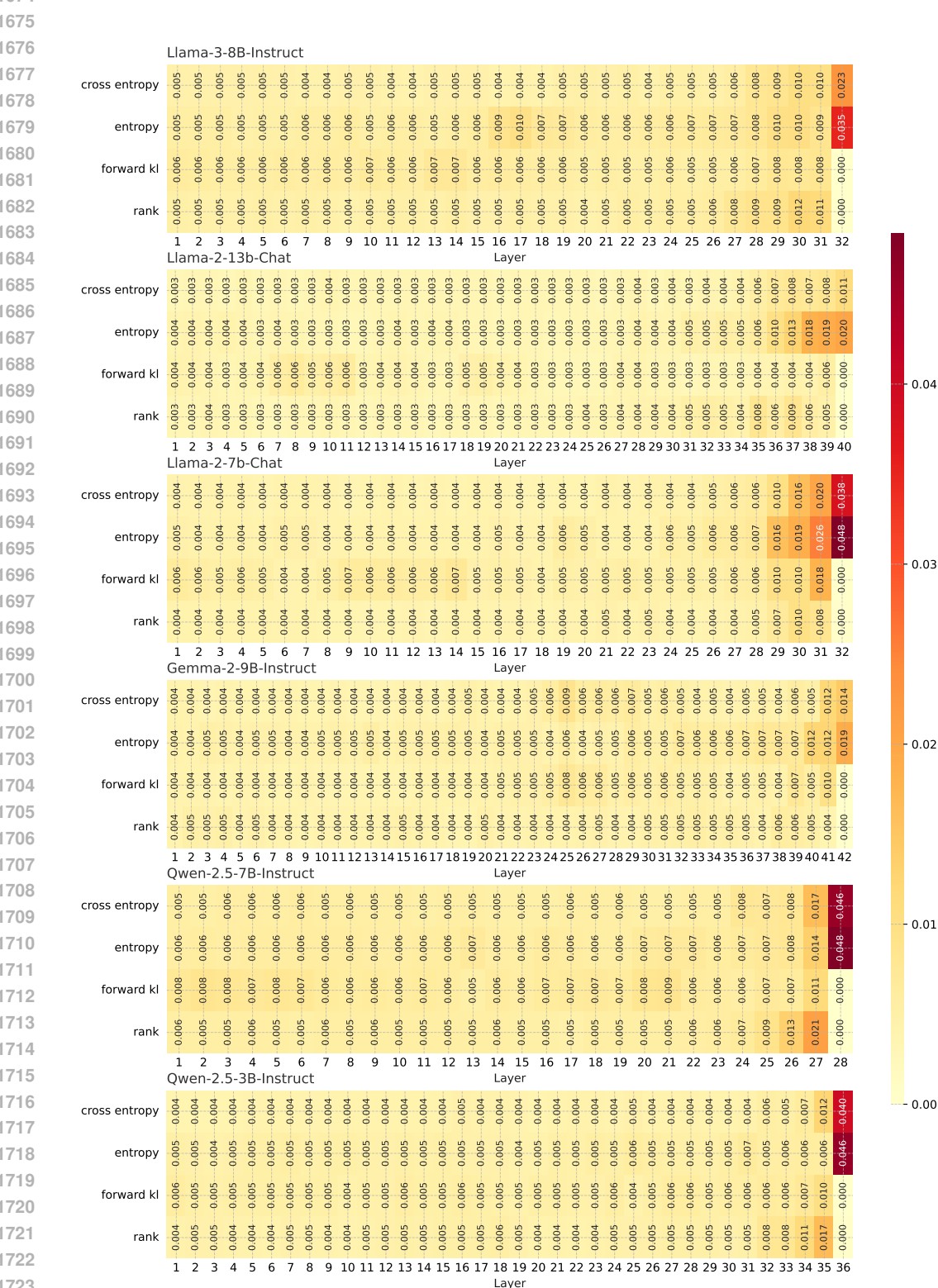

Figure 8: Impurity-based random forest feature importance scores for Logit Lens features from each layer across six models on TriviaQA. Top-$p$ features contribute minimally and are therefore excluded.

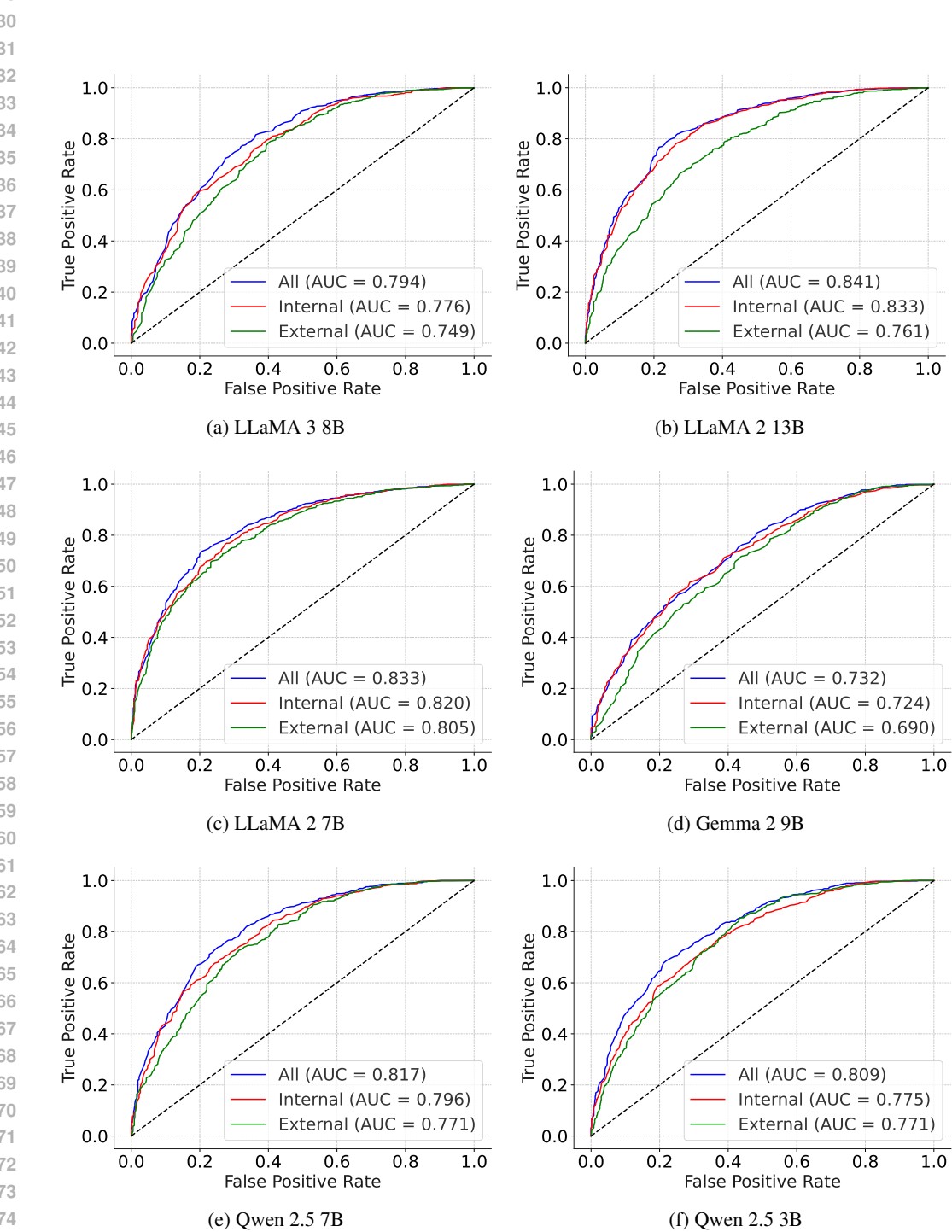

Figure 9: ROC curves comparing classifiers trained on last-layer features (external) versus internal-layer features, across six models on TriviaQA.

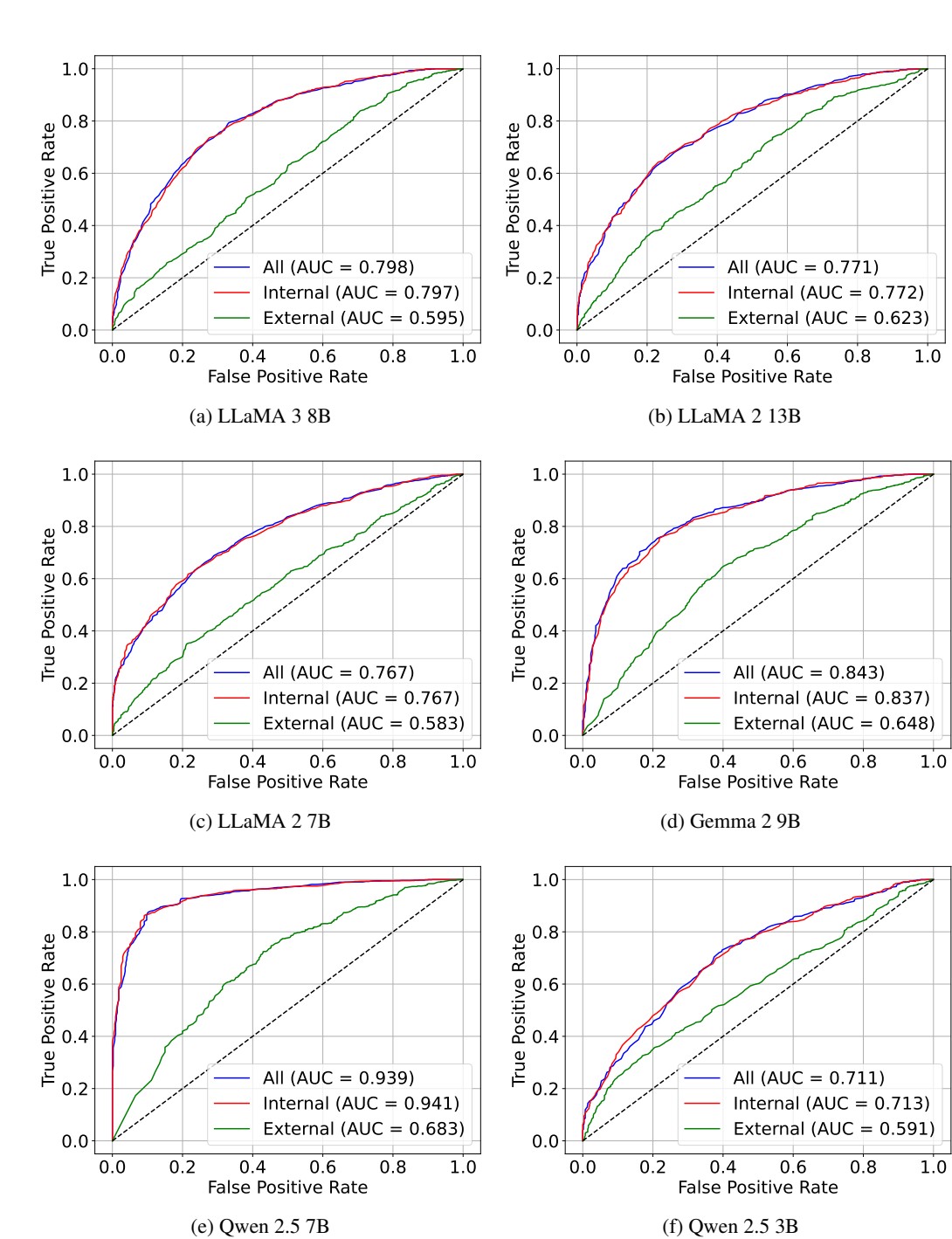

Figure 10: ROC curves comparing classifiers trained on last-layer features (external) versus internal-layer features, across six models on MMLU.

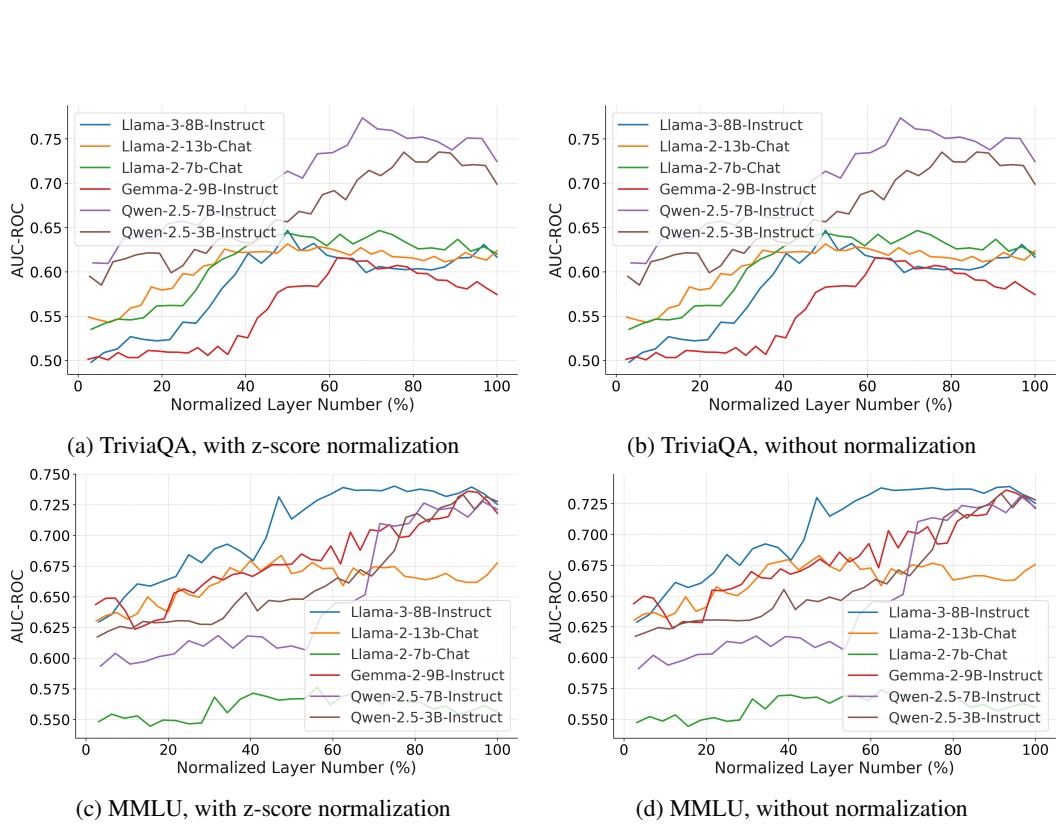

(a) TriviaQA, with z-score normalization

(b) TriviaQA, without normalization

(c) MMLU, with z-score normalization

(d) MMLU, without normalization

Figure 11: Area under ROC curve for random forest classifiers trained on z-score normalized hidden states of each layer. Performance increases with layer depth, suggesting that later layers refine and consolidate decision-relevant signals. Normalization has virtually no effect on performance.

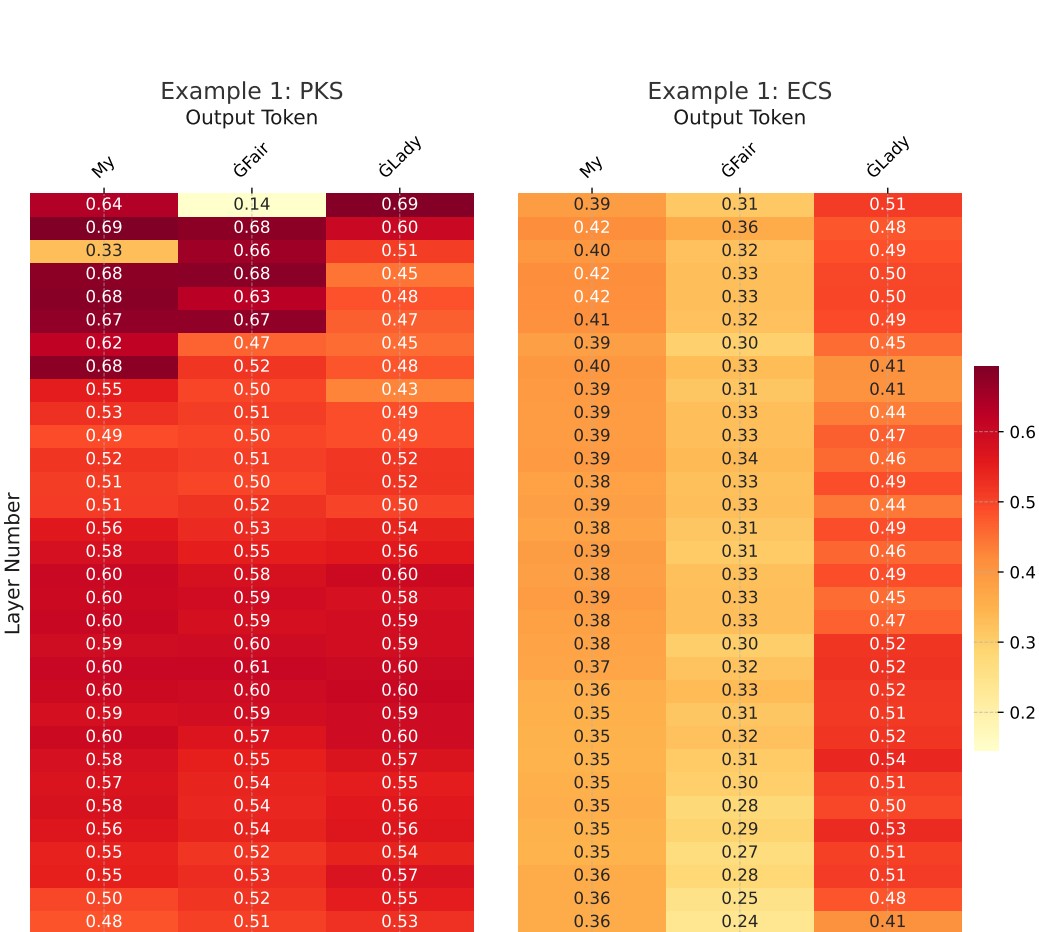

Figure 12: Heatmap example 1 from TriviaQA of PKS and ECS scores for LLaMA-3-8B, illustrating layer-wise token importance. Question: "*Which musical featured the song The Street Where You Live?*" Context: ""*On the Street Where You Live" is a song with music by Frederick Loewe and lyrics by Alan Jay Lerner from the 1956 Broadway musical My Fair Lady. It is sung in the musical by the character Freddy Eynsford-Hill, originally portrayed by John Michael King. In the 1964 film version, the song was performed by Bill Shirley, dubbing for Jeremy Brett. The most popular single was recorded by Vic Damone in 1956, reaching #4 on the Billboard charts and #6 on Cash Box magazine's chart, and it was a #1 hit in the UK in 1958.In 1955, Damone had one song on the charts, "Por Favor," which peaked at #73, but he starred in Hit the Deck and Kismet. In 1956, he moved to Columbia Records, achieving success with hits like "On the Street Where You Live" from My Fair Lady and "An Affair to Remember." His albums on Columbia included That Towering Feeling, Angela Mia, Closer Than a Kiss, This Game of Love, On the Swingin' Side, and Young and Lively. Lyrics describe the narrator's thrill on the street where a loved one lives, highlighting the emotional impact of such proximity. The content is administered by SME and used here for educational purposes under fair use. If concerns arise about unauthorized use, contact the poster. This adheres to the Copyright Act's fair use principles for criticism, comment, news reporting, teaching, scholarship, and research, emphasizing non-profit, educational intentions.*"

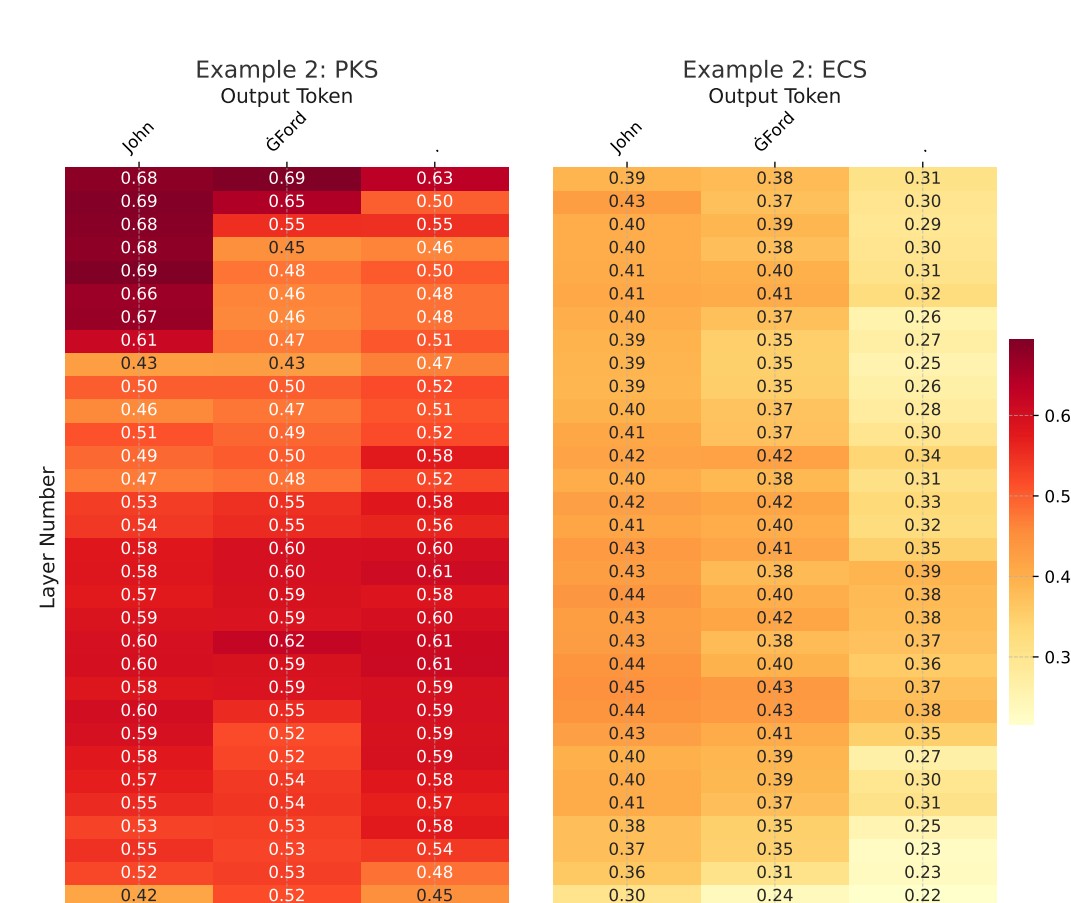

Figure 13: Heatmap example 2 from TriviaQA of PKS and ECS scores for LLaMA-3-8B, illustrating layer-wise token importance. Question: *"Who directed the classic 30s western Stagecoach?"* Context: *"Stagecoach, directed by John Ford (1895-1973), is a quintessential Western notable for its complex characters and Monument Valley setting. John Wayne, portraying "The Ringo Kid," catapulted to stardom in this film. The narrative follows a diverse group of travelers on a stagecoach from Tonto to Lordsburg, facing dangers from an Apache uprising led by Geronimo. Central characters include a falsely accused outlaw Ringo Kid, played by Wayne—seeking to avenge his family's murder—and a prostitute named Dallas, portrayed by Claire Trevor. Other passengers include a drunken doctor (Thomas Mitchell, whose performance won an Academy Award), a gentleman gambler, an embezzling banker, a pregnant army wife, and others, each contributing to a microcosm of society. Ford's direction, coupled with Dudley Nichols and Ben Hecht's script, ensures tight plotting and memorable character arcs, many based on Ernest Haycox's short story "Stage to Lordsburg" and Guy de Maupassant's "Boule de Suif." Ford's handling allowed for minimal screen time yet deep character development. The film features an intense climax with a chase/fight scene that includes Yakima Canutt's pioneering stunts, echoing Spielberg's Raiders of the Lost Ark years later. Stagecoach is celebrated for blending mythic Western landscapes with a poignant social allegory, reflecting on issues like prejudice and redemption. While technical aspects, like some cinematography, show their age, the film remains a paragon from the era, emblematic of Ford's work and establishing recurring Western motifs. It earned seven Oscar nominations, securing wins for Best Supporting Actor and Best Score. Stagecoach's impact persists, marking a pivotal moment in film history and solidifying John Ford's legacy as a master director, influencing numerous directors and films in subsequent years. Whether or not it is perceived as a perfect film, it stands as a significant cultural artifact and vital viewing for any student of cinema."*

1998
1999
2000
2001
2002
2003
2004
2005
2006
2007
2008
2009
2010
2011
2012
2013
2014
2015
2016
2017
2018
2019
2020
2021
2022
2023
2024
2025
2026
2027
2028
2029
2030

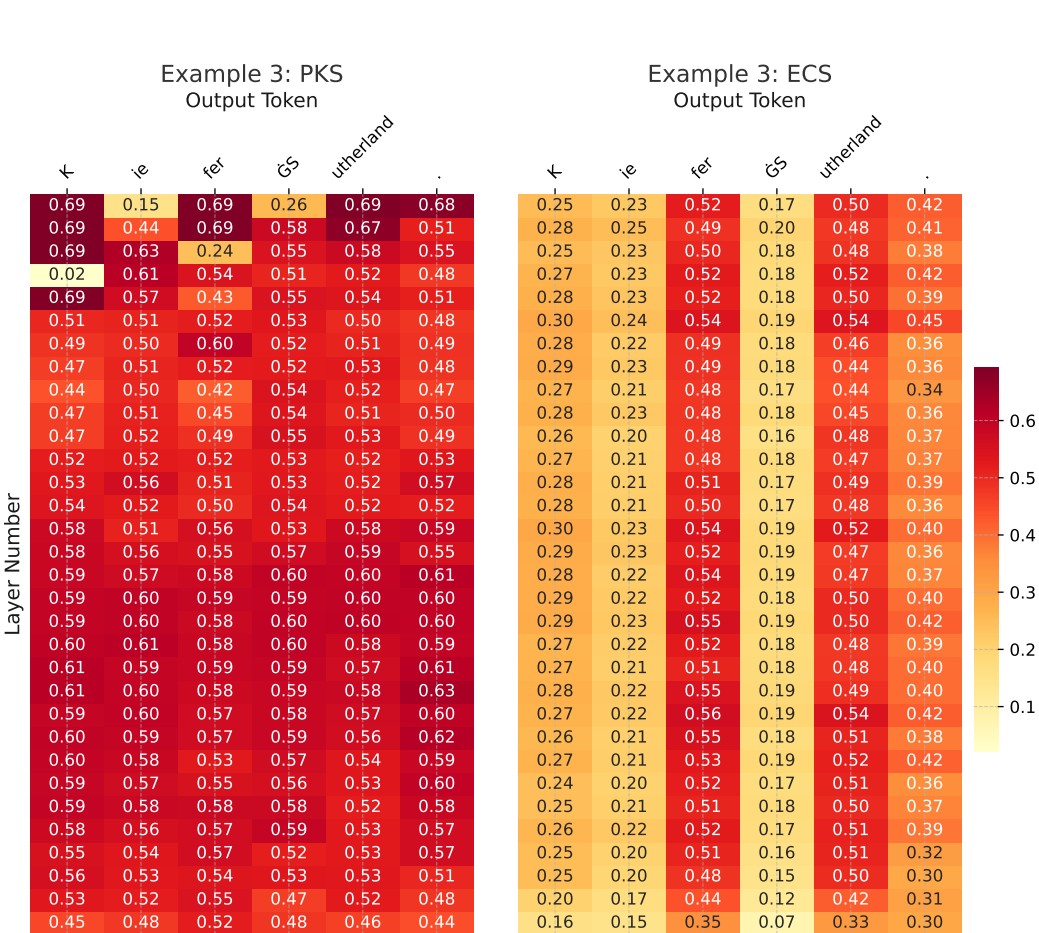

Figure 14: Heatmap example 3 from TriviaQA of PKS and ECS scores for LLaMA-3-8B, illustrating layer-wise token importance. Question: "*Who was born first, Kiefer Sutherland or Christian Slater?*" Context: "*Young Guns II (1990) follows Billy the Kid and his gang as wanted outlaws. The story unfolds with Pat Garrett, a former partner of Billy's, being paid to kill him by cattle baron John Chisum. The movie, directed by Christopher Cain, explores themes of betrayal and redemption as Pat Garrett, who has plans to go respectable, is conflicted about turning against his former friend. Garrett seizes the opportunity to become Sheriff offered by the Governor, who believes in hiring a thief to catch one. This decision to capture Billy opens up great western adventure, with Pat grappling between loyalty and duty. The nuanced performance of William Petersen as Garrett contrasts well with Emilio Estevez reprising his role as the charismatic Billy the Kid. Lou Diamond Phillips elevates his role as Chavez, delivering a performance that is more spiritual and wise than in the first Young Guns film. The talented cast also includes Kiefer Sutherland, Christian Slater, Balthazar Getty, and Alan Ruck. Though depicted as close friends, the real-life association between Pat Garrett and Billy the Kid was less intimate; their familiarity stemmed from mutual patronage of a saloon. Despite this inaccuracy, the film presents an engaging exploration of the dynamics between Garrett and Billy. Young Guns II is a remarkable sequel to 1988's Young Guns, offering a compelling mix of action and moral questions. The film's strong character portrayals, particularly by Estevez and Phillips, enhance its rich narrative of western adventure and friendship against a backdrop of historical myth. It's an exciting film experience that shouldn't be missed.*"
