# OpenReview forum: "LLM Microscope: What Model Internals Reveal About Answer Correctness and Context Utilization"
_ICLR.cc/2026/Conference — Submitted to ICLR 2026_

### Official Review · Reviewer_6JK6 · 2025-10-27

**Soundness:** 3
**Presentation:** 3
**Contribution:** 2
**Rating:** 4
**Confidence:** 4

**Summary:**

This paper investigates whether LLMs internal activations can be used to predict the correctness of generated outputs and to assess the utility of external context. The authors utilize a simple classifiers trained on different types of inputs to predict correctness of the model predictions and show that they can predict correctness with about 75% accuracy. Further, they estimate the efficacy of external context
along w.r.t. correctness and relevancy.

**Strengths:**

Simple and effective approach to generate signal about the correctness in generative tasks.

Introducing new metric for external context validation based on internal states.

The methodology is carefully described and clear.

**Weaknesses:**

I think table 1 adds nothing useful. Besides, to me it seems obvious that LogitLens, TunedLens, and HiddenStates should have close performance since they are all different transformation of the hidden states. Also since there is no specific preference between these three maybe you could just keep one and bring the others in the Appendix as further exploration of different settings.

Please consider modifying the axes ticks and labels in Fig 3 to 6

**Questions:**

I was wondering whether it is possible to directly using the FFN or Attn outputs instead of HiddenStates or PKS etc.

Did you analyse the sensitivity of the results to the parameter $\lambda$?

Did you try more complex classifiers to check if you can further improve the truthfulness detection?

---

> ### Author Response · Authors · 2025-11-27
>
> Thank you for the feedback and for your recognition of our methodology, clarity of presentation, and contributions. We address your questions below:
>
> > 1. I think table 1 adds nothing useful. Besides, to me it seems obvious that LogitLens, TunedLens, and HiddenStates should have close performance since they are all different transformation of the hidden states.
>
> We respectfully disagree that the performance similarity is obvious. In interpretability research, it is not generally safe to assume that transformations such as LogitLens or TunedLens preserve the same level or type of signal as raw hidden states.
>
> - LogitLens projects hidden states through the unembedding matrix, which can distort intermediate representations.
> - TunedLens exists precisely because LogitLens can misalign intermediate logits; the affine corrections explicitly reshape representations.
> - Hidden states preserve high-dimensional relational structure that lens-based projections suppress.
>
> Thus, the fact that these transformations yield similar correctness-prediction performance is a non-trivial empirical finding. Table 1 demonstrates that correctness information is robustly present across many representational spaces, and that even raw hidden states are predictive as well. This supports our claim that correctness signals are widely distributed and accessible early in the computation, which would not be evident without the comparison provided in Table 1.
>
>
> > 2. I was wondering whether it is possible to directly using the FFN or Attn outputs instead of HiddenStates or PKS etc.
>
> Yes, this is feasible. Hidden states are already the aggregated sum of (residual stream) + (FFN output) + (attention output), so they contain both FFN and attention information. Using the components separately would be interesting, and we will discuss the decomposition results in the revision.
>
> > 3. Did you analyse the sensitivity of the results to the parameter $\lambda$?
>
> Yes. Our default $\lambda$ rescales PKS so its magnitude matches ECS, which is the standard and most intuitive normalization. We evaluated the extreme case $\lambda$ = 0 (i.e., ignoring PKS entirely). As shown below, performance drops slightly on both context-differentiation tasks. Using other arbitrary $\lambda$ values would mix PKS and ECS on incompatible scales.
>
> |                                | llama3_8b | llama2_7b | qwen25_7b | qwen25_3b |
> | ------------------------------ | --------- | --------- | --------- | --------- |
> | correct vs. incorrect in Table 3        |     85.5 |     70.2 |     85.2 |     75.9 |
> | correct vs. incorrect $\lambda=0$       |     85.1 |     69.2 |     85.2 |     75.8 |
> | correct vs. irrelevant in Table 3       |     86.3 |     90.5 |     89.6 |     70.6 |
> | correct vs. irrelevant $\lambda=0$      |     84.3 |     89.4 |     89.6 |     70.6 |

---

> ### Author Response · Authors · 2025-11-27
>
> > 4. Did you try more complex classifiers to check if you can further improve the truthfulness detection?
>
> Yes. During the rebuttal period, we evaluated an MLP classifier across all models and datasets. The MLP uses two hidden layers (256 and 64 units), ReLU activations, the Adam optimizer with adaptive learning rates, L2 regularization (α = 0.001), and early stopping. Its performance is consistently on par with or in several cases better than the random forest classifier.
>
> Full results for both TriviaQA and MMLU are included below.
>
>
> **TriviaQA results**:
>
> | Method         | Source  | Metric  | llama3_8b | llama2_13b | llama2_7b | gemma2_9b | qwen25_7b | qwen25_3b |
> | -------------- | ------- | ------- | --------- | ---------- | --------- | --------- | --------- | --------- |
> | **Logit Lens** | Table-2 | ACC     | 0.782     | 0.779      | 0.776     | 0.774     | 0.751     | 0.729     |
> | **Logit Lens** | Table-2 | AUC-ROC | 0.790     | 0.847      | 0.835     | 0.747     | 0.826     | 0.812     |
> | **Logit Lens** | MLP     | ACC     | 0.788     | 0.778      | 0.760     | 0.770     | 0.711     | 0.686     |
> | **Logit Lens** | MLP     | AUC-ROC | 0.790     | 0.844      | 0.820     | 0.758     | 0.764     | 0.755     |
> | **Tuned Lens** | Table-2 | ACC     | 0.779     | 0.775      | 0.775     | -         | -         | -         |
> | **Tuned Lens** | Table-2 | AUC-ROC | 0.782     | 0.846      | 0.829     | -         | -         | -         |
> | **Tuned Lens** | MLP     | ACC     | 0.780     | 0.770      | 0.748     | -         | -         | -         |
> | **Tuned Lens** | MLP     | AUC-ROC | 0.784     | 0.827      | 0.808     | -         | -         | -         |
> | **Hidden States** | Table-2 | ACC     | 0.759     | 0.679      | 0.702     | 0.785     | 0.782     | 0.749     |
> | **Hidden States** | Table-2 | AUC-ROC | 0.647     | 0.631      | 0.647     | 0.616     | 0.774     | 0.735     |
> | **Hidden States** | MLP     | ACC     | 0.760     | 0.681      | 0.704     | 0.806     | 0.772     | 0.757     |
> | **Hidden States** | MLP     | AUC-ROC | 0.729     | 0.655      | 0.681     | 0.722     | 0.770     | 0.752     |
> | **PKS**        | Table-2 | ACC     | 0.733     | 0.725      | 0.709     | 0.743     | 0.650     | 0.695     |
> | **PKS**        | Table-2 | AUC-ROC | 0.729     | 0.768      | 0.743     | 0.723     | 0.715     | 0.752     |
> | **PKS**        | MLP     | ACC     | 0.728     | 0.691      | 0.675     | 0.729     | 0.667     | 0.658     |
> | **PKS**        | MLP     | AUC-ROC | 0.715     | 0.744      | 0.722     | 0.595     | 0.731     | 0.730     |
>
> **MMLU results**:
>
> | Method         | Source  | Metric  | llama3_8b | llama2_13b | llama2_7b | gemma2_9b | qwen25_7b | qwen25_3b |
> | -------------- | ------- | ------- | --------- | ---------- | --------- | --------- | --------- | --------- |
> | **Logit Lens** | Table-2 | ACC     | 0.705     | 0.692      | 0.699     | 0.769     | 0.815     | 0.673     |
> | **Logit Lens** | Table-2 | AUC-ROC | 0.798     | 0.771      | 0.767     | 0.843     | 0.939     | 0.711     |
> | **Logit Lens** | MLP     | ACC     | 0.901     | 0.858      | 0.834     | 0.940     | 0.974     | 0.729     |
> | **Logit Lens** | MLP     | AUC-ROC | 0.961     | 0.933      | 0.914     | 0.975     | 0.990     | 0.796     |
> | **Tuned Lens** | Table-2 | ACC     | 0.705     | 0.692      | 0.699     | -         | -         | -         |
> | **Tuned Lens** | Table-2 | AUC-ROC | 0.798     | 0.771      | 0.767     | -         | -         | -         |
> | **Tuned Lens** | MLP     | ACC     | 0.901     | 0.858      | 0.834     | -         | -         | -         |
> | **Tuned Lens** | MLP     | AUC-ROC | 0.961     | 0.933      | 0.914     | -         | -         | -         |
> | **Hidden States** | Table-2 | ACC     | 0.744     | 0.686      | 0.672     | 0.801     | 0.777     | 0.746     |
> | **Hidden States** | Table-2 | AUC-ROC | 0.740     | 0.684      | 0.576     | 0.736     | 0.728     | 0.733     |
> | **Hidden States** | MLP     | ACC     | 0.748     | 0.679      | 0.664     | 0.779     | 0.770     | 0.730     |
> | **Hidden States** | MLP     | AUC-ROC | 0.753     | 0.676      | 0.585     | 0.736     | 0.741     | 0.723     |
> | **PKS**        | Table-2 | ACC     | 0.605     | -          | 0.543     | 0.705     | 0.691     | 0.618     |
> | **PKS**        | Table-2 | AUC-ROC | 0.537     | -          | 0.543     | 0.555     | 0.541     | 0.538     |
> | **PKS**        | MLP     | ACC     | 0.596     | -          | 0.530     | 0.709     | 0.694     | 0.617     |
> | **PKS**        | MLP     | AUC-ROC | 0.524     | -          | 0.518     | 0.507     | 0.534     | 0.546     |
>
> We hope our responses address your concerns. If you have any further questions, clarifications, or thoughts, we would be happy to answer them. If we have addressed your concerns, we would appreciate updating your score to reflect that.

---

> > ### Comment · Reviewer_6JK6 · 2025-11-27
> >
> > Thank you for your clarifications. I have updated my score and wouldn't mind having the paper accepted, provided that the authors address the responses mentioned above in the camera-ready version.

---

> > > ### Author Response · Authors · 2025-11-27
> > >
> > > Thank you for raising the score! Your feedback is very valuable for improving our paper. We truly appreciate your time and thoughtful comments.

---

### Official Review · Reviewer_3wuj · 2025-10-31

**Soundness:** 3
**Presentation:** 3
**Contribution:** 2
**Rating:** 4
**Confidence:** 4

**Summary:**

The paper develops techniques to use model internals (hidden states, activations, etc.) to predict if a model’s answers are correct and assess the utility of retrieval-augmented context. They study these techniques on multiple different models and show their approach outperforms prompted evaluations.

**Strengths:**

The paper is well-written and develops techniques that improve on prompted baselines to assess answer correctness and context utility. They apply this technique to several different small open-source models for TriviaQA and MMLU.

**Weaknesses:**

While developing calibrated measures of model truthfulness is a critical space, there are multiple existing works that train classifiers on top of different model internals to assess answer correctness. It is unclear if this paper is adding something truly novel to the large existing body of work in this space. Several of these papers are referenced in the related work, but this paper does not benchmark their proposed method against these prior works. Additionally, although the baseline of prompting models to evaluate their correctness may not work well for the model sizes evaluated in the paper (<=13B), prompting alone may be sufficient for larger models (> 32B). Also, it is unclear if new classifiers need to be trained for each model, dataset pair to assess correctness – if so this may reduce the adoption of this technique. Overall the paper is suggesting a method that may not be too distinct from prior works and might not be relevant for models beyond a given size.

**Questions:**

- section 2: Some more related works: Language Models Can Predict Their Own Behavior (https://arxiv.org/pdf/2502.13329), LLM-Check: Investigating Detection of Hallucinations in Large Language Models (https://proceedings.neurips.cc/paper_files/paper/2024/file/3c1e1fdf305195cd620c118aaa9717ad-Paper-Conference.pdf), Are the Hidden States Hiding Something? Testing the Limits of Factuality-Encoding Capabilities in LLMs (https://aclanthology.org/2025.acl-long.304.pdf)

- section 2: Your related work mentions several other “open-box internal-state approaches”. Beyond simple prompting techniques it would be good to include a few of these as baselines.

- section 3.3 line 161: why are the hidden states taken from only the first token position?

- section 3.4 line 168: why does confidence correlate with parametric knowledge used – provide more intuitions?

- section 4.2 line 239-240: strong alignment between the context tokens and generated answer doesn’t necessarily mean that the model is relying more on the context than on parametric knowledge – parametric knowledge itself may also align with the answer

- section 5 line 264: demonstrating this approach on MMLU and TriviaQA is a good starting point – adding 1-2 more datasets being more actively used in current literature such as GPQA would make the results more impactful.

- section 5 line 341: does a separate classifier need to be trained for every model / dataset pair? Can we train a single classifier that could work across several datasets for a given model? Doing so could make this technique more useful.

- section 5 line 322: Other works have shown that larger language models tend to be more calibrated than smaller language models. It’s possible that your prompting baselines may work well for larger models even though they don’t work for your scale. Experimenting with a larger model size (e.g. 32B+) could clarify this.

- section 6.1 line 398: why is PKS predictive of correctness for open-ended QA and not MCQ?

- section 8: Is there a single method that works well across models and datasets for assessing answer correctness and context utility? Explicitly noting this would be valuable for practitioners.

---

> ### Author Response · Authors · 2025-11-27
>
> Thank you for the detailed feedback. We appreciate your recognition of the clarity of our writing, our evaluation across multiple models and datasets, and the practicality of predicting correctness and context utility from internal activations. We address each of your concerns below.
>
> > 1. While developing calibrated measures of model truthfulness is a critical space, there are multiple existing works that train classifiers on top of different model internals to assess answer correctness. It is unclear if this paper is adding something truly novel to the large existing body of work in this space. Several of these papers are referenced in the related work, but this paper does not benchmark their proposed method against these prior works.
>
> Thank you for raising this. To our knowledge, the relevant works cited in our related work, including [1–4], operate under fundamentally different task settings, which makes them unsuitable as direct baselines for our study. In particular:
>
> [1] investigates RQ1 under a simplified and off-policy setting. They use the same hidden states to classify externally provided true/false statements, where the statements are not generated by the same model whose internals are examined. Our work instead evaluates on-policy correctness prediction: whether a model’s internal activations can predict the correctness of its own generated answers.
>
> [2] introduces an unsupervised method that identifies a linear “truth direction” in hidden states. We think of this as a general latent space geometry type finding, one that is not easily directly applicable in our setting for supervised correctness prediction.
>
> [3] trains a P(IK) value head, a single-layer MLP on final-layer activations. Our work has the same setting, but both extends this to all intermediate layers and uses a more interpretable random forest classifier to perform layerwise analysis.
>
> [4] trains a one-layer MLP on hidden states, following the same setup as [3] and our hidden states setting, and is therefore subsumed by our experiments.
>
> Taken together, these prior works either study a different problem formulation or are already included (or exceeded) by the methods we evaluate. This is why we do not benchmark these methods directly in the main paper.
>
>
> > 2. Additionally, although the baseline of prompting models to evaluate their correctness may not work well for the model sizes evaluated in the paper (<=13B), prompting alone may be sufficient for larger models (> 32B).
>
> We agree that larger models may exhibit better self-evaluation accuracy. Unfortunately, due to computational constraints, evaluating 32B–70B-scale models was not feasible during the rebuttal period. We have added this into the limitation section.
>
> > 3. It is unclear if new classifiers need to be trained for each model, dataset pair to assess correctness – if so this may reduce the adoption of this technique.
>
> Thank you for raising this. The two benchmarks we study, TriviaQA and MMLU, represent different tasks (open-ended generation vs. MCQ classification), and consequently the hidden-state statistics associated with their outputs differ. This is analogous to the fact that a model fine-tuned on TriviaQA will not perform well on MMLU without retraining. The same intuition applies here.
>
> Regarding model-specific classifiers: since each model has different architectures, numbers of layers, and hidden-state distributions, transferring a classifier across models is not feasible. We emphasize, however, that the method is identical across all models and the classifiers are extremely lightweight (training takes <1 minute on CPU for all datasets and all models). Thus, while new classifiers are indeed trained per model, we do not view this as a practical limitation.
>
> > 4. Some more related works [5-7]. Your related work mentions several other “open-box internal-state approaches”. Beyond simple prompting techniques it would be good to include a few of these as baselines.
>
> We address the additional works you pointed out:
>
> [5] again trains a linear classifier on hidden states, similar to [3–4], and thus falls under the family of methods we already consider.
>
> [6] computes log-det/entropy-based anomaly scores after the entire output sequence is generated, using teacher-forced full-sequence activations. Our RQ1 instead predicts correctness at the moment the model begins generating, making the methods incomparable in purpose and timing.
>
> [7] is a faithful reproduction of [1] without methodological extensions.
>
> Regarding the “open-box” approaches cited in our related work:
>
> - Logit-based uncertainty baselines are included in our experiments using last-layer logits (e.g., entropy). Figures 9–10 show that such external signals consistently underperform internal feature–based methods.
>
> - Hidden-state probing baselines are already represented in our study through [1–4], which we discuss extensively.
>
> We will clarify this more explicitly in the related work section.

---

> ### Author Response · Authors · 2025-11-27
>
> > 5. Why are the hidden states taken from only the first token position?
>
> Thank you for pointing this out. During the rebuttal period, we ran ablations comparing first-output-token features, versus averages over all generated tokens for all internal-feature methods (LogitLens, TunedLens, Hidden States, PKS) across all models for RQ1. We find that performance remains largely unchanged with token averaging, except that logit lens and tuned lens achieve better results on MMLU. This suggests that averaging across all output tokens may offer some gains; however, using only the first token enables early auditing without requiring the model to generate a full response.
>
> Below we report the averaged vs. first-token results for both TriviaQA and MMLU.
>
> **TriviaQA**:
>
> | Logit lens                   | llama3_8b | llama2_13b | llama2_7b | gemma2_9b | qwen25_7b | qwen25_3b |
> | ------------------------- | --------- | ---------- | --------- | --------- | --------- | --------- |
> | **Averaged – ACC**     | 0.778     | 0.710      | 0.729     | 0.770     | 0.737     | 0.734     |
> | **Averaged – AUC-ROC** | 0.774     | 0.772      | 0.804     | 0.741     | 0.790     | 0.803     |
> | **First-token – ACC**        | 0.782     | 0.779      | 0.776     | 0.774     | 0.751     | 0.729     |
> | **First-token – AUC-ROC**    | 0.790     | 0.847      | 0.835     | 0.747     | 0.826     | 0.812     |
>
> | Tuned lens                  | llama3_8b | llama2_13b | llama2_7b |
> | ------------------------- | --------- | ---------- | --------- |
> | **Averaged – ACC**     | 0.780     | 0.703      | 0.736     |
> | **Averaged – AUC-ROC** | 0.777     | 0.768      | 0.808     |
> | **First-token – ACC**        | 0.779     | 0.775      | 0.775     |
> | **First-token – AUC-ROC**    | 0.782     | 0.846      | 0.829     |
>
> | Hidden States                  | llama3_8b | llama2_13b | llama2_7b | gemma2_9b | qwen25_7b | qwen25_3b |
> | ------------------------- | --------- | ---------- | --------- | --------- | --------- | --------- |
> | **Averaged – ACC**     | 0.739     | 0.714      | 0.707     | 0.773     | 0.782     | 0.768     |
> | **Averaged – AUC-ROC** | 0.611     | 0.670      | 0.633     | 0.584     | 0.769     | 0.754     |
> | **First-token – ACC**        | 0.759     | 0.679      | 0.702     | 0.785     | 0.782     | 0.749     |
> | **First-token – AUC-ROC**    | 0.647     | 0.631      | 0.647     | 0.616     | 0.774     | 0.735     |
>
> | PKS                   | llama3_8b | llama2_7b | gemma2_9b | qwen25_7b | qwen25_3b |
> | ------------------------- | --------- | --------- | --------- | --------- | --------- |
> | **Averaged – ACC**        | 0.733     | 0.709     | 0.743     | 0.650     | 0.695     |
> | **Averaged – AUC-ROC**    | 0.729     | 0.743     | 0.723     | 0.715     | 0.752     |
> | **First-token – ACC**     | 0.737     | 0.668     | 0.733     | 0.659     | 0.684     |
> | **First-token – AUC-ROC** | 0.740     | 0.741     | 0.642     | 0.702     | 0.731     |
>
> **MMLU**:
>
> | Logit lens                   | llama3_8b | llama2_13b | llama2_7b | gemma2_9b | qwen25_7b | qwen25_3b |
> | ------------------------- | --------- | ---------- | --------- | --------- | --------- | --------- |
> | **Averaged – ACC**     | 0.836     | 0.717      | 0.684     | 0.814     | 0.822     | 0.803     |
> | **Averaged – AUC-ROC** | 0.941     | 0.789      | 0.757     | 0.904     | 0.945     | 0.908     |
> | **First-token – ACC**        | 0.705     | 0.692      | 0.699     | 0.769     | 0.815     | 0.673     |
> | **First-token – AUC-ROC**    | 0.798     | 0.771      | 0.767     | 0.843     | 0.939     | 0.711     |
>
> | Tuned lens                   | llama3_8b | llama2_13b | llama2_7b |
> | ------------------------- | --------- | ---------- | --------- |
> | **Averaged – ACC**     | 0.826     | 0.728      | 0.680     |
> | **Averaged – AUC-ROC** | 0.928     | 0.815      | 0.768     |
> | **First-token – ACC**        | 0.695     | 0.671      | 0.684     |
> | **First-token – AUC-ROC**    | 0.770     | 0.752      | 0.760     |
>
> | Hidden States                  | llama3_8b | llama2_13b | llama2_7b | gemma2_9b | qwen25_7b | qwen25_3b |
> | ------------------------- | --------- | ---------- | --------- | --------- | --------- | --------- |
> | **Averaged – ACC**     | 0.745     | 0.740      | 0.698     | 0.798     | 0.780     | 0.745     |
> | **Averaged – AUC-ROC** | 0.741     | 0.737      | 0.613     | 0.739     | 0.730     | 0.732     |
> | **First-token – ACC**        | 0.744     | 0.686      | 0.672     | 0.801     | 0.777     | 0.746     |
> | **First-token – AUC-ROC**    | 0.740     | 0.684      | 0.576     | 0.736     | 0.728     | 0.733     |

---

> ### Author Response · Authors · 2025-11-27
>
> | PKS                   | llama3_8b | llama2_7b | gemma2_9b | qwen25_7b | qwen25_3b |
> | ------------------------- | --------- | --------- | --------- | --------- | --------- |
> | **Averaged – ACC**        | 0.605     | 0.543     | 0.705     | 0.691     | 0.618     |
> | **Averaged – AUC-ROC**    | 0.537     | 0.543     | 0.555     | 0.541     | 0.538     |
> | **First-token – ACC**     | 0.588     | 0.529     | 0.708     | 0.694     | 0.615     |
> | **First-token – AUC-ROC** | 0.524     | 0.510     | 0.526     | 0.565     | 0.542     |
>
>
> > 6. Why does confidence correlate with parametric knowledge used – provide more intuitions?
>
> Intuitively: a low PKS indicates the FFN layers contribute little parametric knowledge to the predicted answer. Our empirical findings suggest the model may not know enough about the topic when it exhibits this low PKS score, which correlates with lower output confidence and uncertainty in general.
>
>
> > 7. Strong alignment between the context tokens and generated answer doesn’t necessarily mean that the model is relying more on the context than on parametric knowledge – parametric knowledge itself may also align with the answer
>
> We agree with your observation, and this is precisely why PKS is included in our RQ2 formulation. Our statement in lines 239–240: “a higher ECS indicates stronger alignment with the retrieved context”, does not claim that the model relies more on context than on parametric knowledge. It only states that high ECS implies the context is being effectively utilized. PKS is included to disambiguate reliance on parametric memory.
>
> > 8. Demonstrating this approach on MMLU and TriviaQA is a good starting point – adding 1-2 more datasets being more actively used in current literature such as GPQA would make the results more impactful.
>
> GPQA contains only 198 diamond and 448 main examples, which is too small. We excluded it for this reason.
>
> > 9. Why is PKS predictive of correctness for open-ended QA and not MCQ?
>
> This is a good question. In open-ended QA (e.g., TriviaQA), the model must generate a factual token such as an entity, so the FFN layers tend to inject parametric knowledge directly into the answer representation, producing a strong PKS signal.
>
> In MCQ (MMLU), the final output token is simply “A/B/C/D”, which does not itself encode factual content. As a result, FFN layers may not provide distinguishable knowledge signals at that output position, and PKS becomes uninformative.
>
> We hope our responses address your concerns. If you have any further questions, clarifications, or thoughts, we would be happy to answer them. If we have addressed your concerns, we would appreciate updating your score to reflect that.
>
> [1] The internal state of an LLM knows when it‘s lying
>
> [2] Discovering latent knowledge in language models without supervision
>
> [3] Language models (mostly) know what they know
>
> [4] Llms know more than they show: On the intrinsic representation of llm hallucinations
>
> [5] Language Models Can Predict Their Own Behavior
>
> [6] LLM-Check: Investigating Detection of Hallucinations in Large Language Models
>
> [7] Are the Hidden States Hiding Something? Testing the Limits of Factuality-Encoding Capabilities in LLMs

---

> > ### Author Response · Authors · 2025-11-28
> >
> > Dear reviewer, we would like to kindly ask if we have addressed your concerns. We sincerely appreciate your questions and feedback!

---

### Official Review · Reviewer_Q3yN · 2025-11-01

**Soundness:** 3
**Presentation:** 3
**Contribution:** 3
**Rating:** 6
**Confidence:** 3

**Summary:**

The paper proposes LLM Microscope to predict whether a language models answer will be correct using internal activations alone and whether external content is helpful or harmful. For the first hypothesis they train light weight classifier over features derived from intermediate representations Logit-Lens statistics, first-token hidden states, and a Parametric Knowledge Score.
For the second hypothesis (regarding the external content) they introduce an internals based proxy for contextual log-likelihood gain by combining an External Context Score via a scaling parameter.
The authors show experiment across multiple models and show correctness can be predicted via internals at 75% accuracy and a high AUC, and internal based signals outperform prompting when distinguishing correct vs incorrect context.

**Strengths:**

1) The paper clearly formulates the problem and cleanly formulates contextual log likelihood gain and relative utility and instanties an internal based proxy for context efficacy. Definitions and rationale are explicit.
2) Random forests over per-layer logit lens statistics and first token hidden states is a well setup experiment and yield strong correctness predictions with analysis of feature importance and layer-wise trends. The observation that internal layers are as or more predictive than the final layer is compelling and practically useful for early auditing.
3) The authors perform extensive experiments that covers 6 models. Table 2 is important and shows internals-based features consistently outperform prompting baselines.

**Weaknesses:**

1) So the “Incorrect” contexts are produced by prompting GPT-4o to replace mentions of the gold answer with plausible but wrong alternatives, whereas “correct” contexts are sometimes GPT-4o-summarized to 500 tokens. This could introduce stylistic artifacts that models might pick up rather than genuine semantic correction signals. More diagnostics are needed to show that performance improvements are not due to such artifacts, such analysis is missing in the paper and is critical.
2) Many generative answers extend beyond a single token and while the paper acknowledges this in Limitations, most RQ1 features are computed at the first output token and some RQ2 averages are token-averaged but layer-averaged later.

**Questions:**

1) There is some inconsistency in the TriviaQA counts, in the experimental setup is states it retains 6557 questions but in D.1 the paper says quality filter yields 11683 examples.

---

> ### Author Response · Authors · 2025-11-27
>
> Thank you for the thoughtful review and for recognizing the strengths of our work, including the clear formulation of the problem, the design of the correctness-prediction experiments, the insight that intermediate layers are often more predictive than the final layer, and the breadth of our evaluation across six models. We address your concerns below.
>
> > 1. Concern about stylistic artifacts in correct vs. incorrect contexts
>
> There may be a misunderstanding about how the correct and incorrect contexts are constructed. Both correct and incorrect contexts are from the same GPT-4o-summarized passage. To create an incorrect context, we perform a minimal edit: we replace only the gold answer spans with a plausible but wrong alternative, leaving all other text unchanged.
>
> Thus, there is no difference in summarization procedure or passage structure, and the only modification is a localized substitution of the answer content itself. To verify this empirically, during the rebuttal period we manually inspected 100 randomly sampled correct-incorrect context pairs. We did not observe stylistic artifacts or systematic surface-level cues beyond the intended substitution. We have clarified this procedure to avoid confusion in Line 280.
>
>
>
> > 2. Concern about using only the first output token for RQ1 and averaging choices in RQ2
>
> Thank you for pointing this out. During the rebuttal period, we ran ablations comparing first-output-token features, versus averages over all generated tokens for all internal-feature methods (LogitLens, TunedLens, PKS, Hidden States) across all models for RQ1. We find that performance remains largely unchanged with token averaging, except that logit lens and tuned lens achieve better results on MMLU. This suggests that averaging across all output tokens may offer some gains; however, using only the first token enables early auditing without requiring the model to generate a full response.
>
> Below we report the averaged vs. first-token results for RQ1 for both TriviaQA and MMLU.

---

> ### Author Response · Authors · 2025-11-27
>
> **TriviaQA**:
>
> | Logit lens                   | llama3_8b | llama2_13b | llama2_7b | gemma2_9b | qwen25_7b | qwen25_3b |
> | ------------------------- | --------- | ---------- | --------- | --------- | --------- | --------- |
> | **Averaged – ACC**     | 0.778     | 0.710      | 0.729     | 0.770     | 0.737     | 0.734     |
> | **Averaged – AUC-ROC** | 0.774     | 0.772      | 0.804     | 0.741     | 0.790     | 0.803     |
> | **First-token – ACC**        | 0.782     | 0.779      | 0.776     | 0.774     | 0.751     | 0.729     |
> | **First-token – AUC-ROC**    | 0.790     | 0.847      | 0.835     | 0.747     | 0.826     | 0.812     |
>
> | Tuned lens                  | llama3_8b | llama2_13b | llama2_7b |
> | ------------------------- | --------- | ---------- | --------- |
> | **Averaged – ACC**     | 0.780     | 0.703      | 0.736     |
> | **Averaged – AUC-ROC** | 0.777     | 0.768      | 0.808     |
> | **First-token – ACC**        | 0.779     | 0.775      | 0.775     |
> | **First-token – AUC-ROC**    | 0.782     | 0.846      | 0.829     |
>
> | Hidden States                  | llama3_8b | llama2_13b | llama2_7b | gemma2_9b | qwen25_7b | qwen25_3b |
> | ------------------------- | --------- | ---------- | --------- | --------- | --------- | --------- |
> | **Averaged – ACC**     | 0.739     | 0.714      | 0.707     | 0.773     | 0.782     | 0.768     |
> | **Averaged – AUC-ROC** | 0.611     | 0.670      | 0.633     | 0.584     | 0.769     | 0.754     |
> | **First-token – ACC**        | 0.759     | 0.679      | 0.702     | 0.785     | 0.782     | 0.749     |
> | **First-token – AUC-ROC**    | 0.647     | 0.631      | 0.647     | 0.616     | 0.774     | 0.735     |
>
> | PKS                   | llama3_8b | llama2_7b | gemma2_9b | qwen25_7b | qwen25_3b |
> | ------------------------- | --------- | --------- | --------- | --------- | --------- |
> | **Averaged – ACC**        | 0.733     | 0.709     | 0.743     | 0.650     | 0.695     |
> | **Averaged – AUC-ROC**    | 0.729     | 0.743     | 0.723     | 0.715     | 0.752     |
> | **First-token – ACC**     | 0.737     | 0.668     | 0.733     | 0.659     | 0.684     |
> | **First-token – AUC-ROC** | 0.740     | 0.741     | 0.642     | 0.702     | 0.731     |
>
> **MMLU**:
>
> | Logit lens                   | llama3_8b | llama2_13b | llama2_7b | gemma2_9b | qwen25_7b | qwen25_3b |
> | ------------------------- | --------- | ---------- | --------- | --------- | --------- | --------- |
> | **Averaged – ACC**     | 0.836     | 0.717      | 0.684     | 0.814     | 0.822     | 0.803     |
> | **Averaged – AUC-ROC** | 0.941     | 0.789      | 0.757     | 0.904     | 0.945     | 0.908     |
> | **First-token – ACC**        | 0.705     | 0.692      | 0.699     | 0.769     | 0.815     | 0.673     |
> | **First-token – AUC-ROC**    | 0.798     | 0.771      | 0.767     | 0.843     | 0.939     | 0.711     |
>
> | Tuned lens                   | llama3_8b | llama2_13b | llama2_7b |
> | ------------------------- | --------- | ---------- | --------- |
> | **Averaged – ACC**     | 0.826     | 0.728      | 0.680     |
> | **Averaged – AUC-ROC** | 0.928     | 0.815      | 0.768     |
> | **First-token – ACC**        | 0.695     | 0.671      | 0.684     |
> | **First-token – AUC-ROC**    | 0.770     | 0.752      | 0.760     |
>
> | Hidden States                  | llama3_8b | llama2_13b | llama2_7b | gemma2_9b | qwen25_7b | qwen25_3b |
> | ------------------------- | --------- | ---------- | --------- | --------- | --------- | --------- |
> | **Averaged – ACC**     | 0.745     | 0.740      | 0.698     | 0.798     | 0.780     | 0.745     |
> | **Averaged – AUC-ROC** | 0.741     | 0.737      | 0.613     | 0.739     | 0.730     | 0.732     |
> | **First-token – ACC**        | 0.744     | 0.686      | 0.672     | 0.801     | 0.777     | 0.746     |
> | **First-token – AUC-ROC**    | 0.740     | 0.684      | 0.576     | 0.736     | 0.728     | 0.733     |
>
> | PKS                   | llama3_8b | llama2_7b | gemma2_9b | qwen25_7b | qwen25_3b |
> | ------------------------- | --------- | --------- | --------- | --------- | --------- |
> | **Averaged – ACC**        | 0.605     | 0.543     | 0.705     | 0.691     | 0.618     |
> | **Averaged – AUC-ROC**    | 0.537     | 0.543     | 0.555     | 0.541     | 0.538     |
> | **First-token – ACC**     | 0.588     | 0.529     | 0.708     | 0.694     | 0.615     |
> | **First-token – AUC-ROC** | 0.524     | 0.510     | 0.526     | 0.565     | 0.542     |
>
> Across nearly all cases, full-token averaging yields similar or slightly better performance to first-token features.
>
>
> > 3. There is some inconsistency in the TriviaQA counts, in the experimental setup it states it retains 6557 questions but in D.1 the paper says quality filter yields 11683 examples.
>
> Thank you for catching this. The correct dataset size used in all experiments is 6557 examples.
> The number 11,683 actually refers to the pool before deduplication. Section D.1 has now been corrected to reflect this.

---

> ### Author Response · Authors · 2025-11-28
>
> Dear reviewer, we would like to kindly ask if we have addressed your concerns. If you have any further questions, clarifications, or thoughts, we would be happy to answer them. If we have addressed your concerns, we would appreciate updating your score to reflect that. We sincerely appreciate your questions and feedback!

---

### Official Review · Reviewer_p55B · 2025-11-04

**Soundness:** 3
**Presentation:** 3
**Contribution:** 2
**Rating:** 4
**Confidence:** 4

**Summary:**

This paper provides an empirical study of how model internal states and other signals can be used for predicting: (i) if the model output is correct; and (ii) how much it relies on the external context. The results show that internal hidden states are effective at predicting correctness, and that features derived from external context matching and FFN activation strength are predictive of context relevance.

**Strengths:**

- The experiments are designed well and important aspects related to understanding LLM behaviors. The results are well presented and show clear trends in support of the claims made in the paper.
- The paper does a good job of bringing together multiple lines of research around mechanistic interpretability, confidence elicitation from LLMs, contextual faithfulness and factuality. The findings presented are useful for driving further research in these areas.

**Weaknesses:**

- All techniques in the paper are borrowed from prior works -- ECS and PKS scores (Sun et al, 2025), predicting factuality from hidden states (Azaria & Mitchell, 2023), verbalized confidence (Kadavath et al, 2022), Logit lens and Tuned lens. While the paper does a good job of contrasting and comparing them, it doesn't make any new methodological contributions.
- Consequently, a lot of the main conclusions in the paper are already well known -- e.g., that hidden states are predictive of factuality, ECS and PKS scores are effective at measuring context vs parametric knowledge utilization.
- Dealing with incorrect / conflicting contexts is a rich area with lots of papers. The paper misses out discussion of some important techniques, e.g., from ICLR 2025 [1].

[1] Huang, Yukun, et al. "To Trust or Not to Trust? Enhancing Large Language Models' Situated Faithfulness to External Contexts." The Thirteenth International Conference on Learning Representations.

**Questions:**

- Can you explain how the findings related to RQ1 here are different from those of Azaria & Mitchell (2023)?

---

> ### Author Response · Authors · 2025-11-27
>
> Thank you for the thoughtful feedback. We are glad that you found value in our experiment design for understanding LLM behaviors, the alignment between our experiments and claims, and our effort to connect multiple lines of existing work while outlining directions for future research. We address your concerns below:
>
> > 1. All techniques in the paper are borrowed from prior works. Also how is RQ1 different from Azaria & Mitchell (2023) [1]?
>
> Thank you for the question. Our contribution lies not only in applying existing interpretability tools but in designing two novel and underexplored tasks (RQ1 and RQ2), evaluated across six models, that have not been systematically studied before. These tasks enable a unified, comparative examination of how different interpretability methods such as LogitLens, ECS, PKS, and hidden-state probing help answer key questions about LLM behavior. While the techniques themselves originate from prior work, our novelty is in using them in new settings, demonstrating their performance and applicability through comprehensive experiments and cross-method comparisons. We explore how interpretability techniques can be used as lenses to build models that predict when a model is correct (RQ1) and whether auxiliary context is relevant and/or correct (RQ2).
>
> Regarding [1], that work studies RQ1 in a simplified and fundamentally different setting. Their task is to classify externally provided true/false statements, where the statements are not generated by the same model whose hidden states are analyzed. In contrast, our study evaluates whether a model’s internals can predict the correctness of its own generations, creating a more direct, on-policy setting. We also extend the comparison of hidden states to logit lens, tuned lens, PKS, and prompting baselines, providing a much broader evaluation than [1].
>
>
> > 2. Consequently, a lot of the main conclusions in the paper are already well known -- e.g., that hidden states are predictive of factuality, ECS and PKS scores are effective at measuring context vs parametric knowledge utilization.
>
> While prior work suggests that hidden states correlate with factuality, the definition and evaluation of factuality prediction in our work differ substantially, as explained above and in the introduction and related work section. We introduce a more direct, online setting where internal signals are extracted from the same model as it generates outputs, rather than in offline or off-policy setups.
>
> Similarly, PKS and ECS have previously been used exclusively for hallucination detection, not for measuring context vs. parametric knowledge utilization, and no prior experiments evaluate their effectiveness for this purpose. In contrast, we systematically analyze (1) whether PKS is useful for RQ1 and (2) how PKS and ECS can explicitly identify context utility. As you noted, our experiments help bridge gaps across prior methods through new tasks, which we believe constitutes an important contribution.
>
>
> > 3. Dealing with incorrect / conflicting contexts is a rich area with lots of papers. The paper misses out discussion of some important techniques, e.g., from ICLR 2025 [2].
>
> Thank you for pointing this out. The referenced ICLR 2025 paper adopts a standard prompting-based method: it elicits confidence estimates via verbalized reasoning to decide whether to trust internal or external information. In contrast, our work uses mechanistic interpretability to directly analyze model internals and compare them against prompting-based baselines, offering insight into how internal computations reflect context utilization. We have added this paper to the related work section.
>
> We hope our responses address your concerns. If you have any further questions, clarifications, or thoughts, we would be happy to answer them. If we have addressed your concerns, we would appreciate updating your score to reflect that.
>
> [1] The Internal State of an LLM Knows When It's Lying
>
> [2] To Trust or Not to Trust? Enhancing Large Language Models' Situated Faithfulness to External Contexts.

---

> > ### Author Response · Authors · 2025-11-28
> >
> > Dear reviewer, we would like to kindly ask if we have addressed your concerns. We sincerely appreciate your questions and feedback!

---

### Meta-Review · Area_Chair_xq31 · 2026-01-06

**Summary:**

This paper studies for RAG LLM, whether model-internal activations can predict 1, the correctness of an LLM’s own generated answer and 2, whether retrieved external context is helpful, harmful, or irrelevant. The authors evaluate a set of internal signals (hidden-state probing, logit lens / tuned lens statistics, activations-based PKS/ECS-style scores) and train lightweight classifiers, reporting that internal features can predict correctness at ~75% accuracy and that an internals-based context utility metric outperforms prompting baselines on distinguishing correct vs incorrect context. Experiments span six open-source models and on TriviaQA and MMLU, with context-efficacy experiments centered on TriviaQA.

Overall, reviewers agree the problem is important and the experiments are clearly presented, but raise significant concerns about novelty, external validity, and practical significance, which remain insufficiently resolved.

Strengths
1. Well-executed empirical study across multiple models, with generally consistent trends and helpful analyses.

Weaknesses

1. Limited novelty: Multiple reviewers argue the paper mainly repackages and compares techniques from prior work (hidden-state probing for truthfulness/factuality, logit/tuned lens, PKS/ECS-like scores, etc.). While the authors position novelty as “new tasks” and “on-policy” correctness prediction, the methodological contribution remains limited.
2. The incorrect-context construction relies on synthetic perturbations (answer-span substitutions within summarized passages). While this is a reasonable controlled setup, it leaves concerns whether the proposed internals-based signals robustly detect real retrieval failure modes. Reviewers requested stronger diagnostics; the rebuttal provides clarification and a small manual inspection, but does not fully de-risk the concern
3. Limited evaluation scope
4. A key adoption concern is whether separate classifiers must be trained per model and potentially per task/dataset. The rebuttal argues training is cheap, but does not demonstrate cross-task generalization or a reusable “single probe” story. This makes the approach less like a broadly deployable primitive.

**Reviewer Concerns:**

Addressed

1. Added systematic ablations comparing first-token features vs token-averaged features across models/methods
2. Ran MLP experiments and reported comparable/better performance in many cases.
3. Provided an extreme-case check (λ=0) and argued scaling rationale.

Still Outstanding

see weakness

**Reviewer Scores:**

6JK6 is likely to increase score to 5
Q3yN/p55B/3wuj may stay at current scores

---

### Decision · Program_Chairs · 2026-01-26

Reject